# Training Transformers with 4-bit Integers

**Haocheng Xi[2*], Changhao Li[1], Jianfei Chen[1†], and Jun Zhu[1]**
[1]Dept. of Comp. Sci. and Tech., Institute for AI, BNRist Center, THBI Lab,
Tsinghua-Bosch Joint ML Center, Tsinghua University
[2]Institute for Interdisciplinary Information Sciences, Tsinghua University
{xihc20,lichangh20}@mails.tsinghua.edu.cn, {jianfeic,dcszj}@tsinghua.edu.cn

## Abstract

Quantizing the activation, weight, and gradient to 4-bit is promising to accelerate neural network training. However, existing 4-bit training methods require custom numerical formats which are not supported by contemporary hardware. In this work, we propose a training method for transformers with all matrix multiplications implemented with the INT4 arithmetic. Training with an ultra-low INT4 precision is challenging. To achieve this, we carefully analyze the specific structures of activation and gradients in transformers to propose dedicated quantizers for them. For forward propagation, we identify the challenge of outliers and propose a Hadamard quantizer to suppress the outliers. For backpropagation, we leverage the structural sparsity of gradients by proposing bit splitting and leverage score sampling techniques to quantize gradients accurately. Our algorithm achieves competitive accuracy on a wide range of tasks including natural language understanding, machine translation, and image classification. Unlike previous 4-bit training methods, our algorithm can be implemented on the current generation of GPUs. Our prototypical linear operator implementation is up to 2.2 times faster than the FP16 counterparts and speeds up the training by 17.8% on average for sufficiently large models. Our code is available at https://github.com/xijiu9/Train_Transformers_with_INT4.

## 1 Introduction

Training neural networks is computationally demanding. Training with low-precision arithmetic (a.k.a., fully quantized training or FQT) is promising to improve computational and memory efficiency. FQT methods add some quantizers and dequantizers in the original full-precision computational graph, and replace expensive floating-point operations with cheap low-precision ones.

Research in FQT aims to reduce the training numerical precision, without sacrificing much convergence speed or accuracy. The required numerical precision has been reduced from FP16 [33] to FP8 [54, 46], INT32+INT8 [3] and INT8+INT5 [7]. FP8 training is implemented in Nvidia's H100 GPU with Transformer Engine [35], achieving impressive speedup for the training of large-scale transformers. Recently, the training numerical precision has been pushed down to 4 bits. Sun et al. [47] successfully trained several modern networks with INT4 activation/weights and FP4 gradients; and Chmiel et al. [8] propose a custom 4-bit logarithmic numerical format to further improve the accuracy. However, these 4-bit training methods cannot be directly utilized for acceleration as they require custom numerical formats that are not supported on contemporary hardware.

There are significant optimization challenges to train neural networks at an extremely low 4-bit level. First, the non-differentiable quantizers in forward propagation make the loss landscape rugged, where gradient-based optimizers can easily stuck at local optima [31]. Second, gradients are only computed

---

[*]This work was done during an internship in the Department of Computer Science and Technology, Tsinghua University

[†]Corresponding author.

37th Conference on Neural Information Processing Systems (NeurIPS 2023).

approximately in low-precision. Such imprecise gradients slow down the training process and even cause the training to be unstable or diverge.

In this work, we propose a novel INT4 training algorithm for a class of popular neural networks, transformers [52]. All the costly linear operations for training transformers can be written in a matrix multiplication (MM) form. This MM form allows us to design more flexible quantizers, which better approximate FP32 matrix multiplications by utilizing specific structures of the activations, weights, and gradients in transformers. Our quantizers leverage advances in the field of randomized numerical linear algebra (RandNLA) [14].

For forward propagation, we find that outliers in the activation are the main reason for accuracy degradation. To suppress outliers, we propose a *Hadamard quantizer*, which quantizes a *transformed version* of the activation matrix. The transformation is a block diagonal Hadamard matrix, which spreads the information carried in outliers to its nearby entries of the matrix and thus reduces the numerical range of the outliers.

For backpropagation, we exploit the *structural sparsity* of activation gradients. We find that the gradients of a few tokens are extremely large. Meanwhile, the gradients for the rest majority of the tokens are very small, even smaller than the quantization residuals of larger gradients. Rather than computing these small gradients, it is better to save the computational resources for calculating the residuals of the larger gradients. To utilize such sparsity, we propose *bit splitting*, which splits the gradient of each token into higher 4 bits and lower 4 bits. Then, we choose the most informative gradients by *leverage score sampling*, which is an importance sampling technique for RandNLA.

Combining quantization techniques for forward and backward propagation, we propose an algorithm that uses INT4 MMs for all linear operations in transformers. We evaluate our algorithm for training transformers on a wide variety of tasks, including natural language understanding, question answering, machine translation, and image classification. Our algorithm achieves competitive or superior accuracy compared with existing works on 4-bit training [47, 8]. Moreover, our algorithm *is compatible with contemporary hardware* like GPUs, since it does not require custom numerical formats like FP4 or logarithm formats. Our prototypical quantization + INT4 MM operator implementation is up to 2.2 times faster than the FP16 MM baseline, and it speeds up the training by up to 35.1%.

Finally, we would like to point out that utilizing ultra-low 4-bit numerical formats for training neural networks is still an open problem, and the main purpose of this research is to study whether it is possible to design an INT4 training algorithm that can achieve reasonable accuracy on practical tasks. The current research state of INT4 training is not yet mature enough to provide significant speedup for most tasks in a generic and plug-and-play manner. See Sec. 6 for further discussions.

## 2 Related Work

**Fully Quantized Training**    Fully quantized training (FQT) [33, 54, 46, 3, 15, 1, 57, 65, 29, 30, 59, 68] methods accelerate training by quantizing the activations, weights, and gradients to low-precision, so linear and nonlinear operators during training can be implemented with low-precision arithmetic. Research on FQT design novel numerical formats and quantization algorithms that better approximate full-precision tensors. The current research frontier is 4-bit FQT. FQT is challenging due to the vast numerical range of the gradient and the optimization issues of training quantized networks from scratch. Due to these challenges, existing 4-bit FQT algorithms [47, 8] still have ∼1-2.5% accuracy drop on several tasks, and they cannot support contemporary hardware.

**Other Efficient Training Methods**    Mixture-of-experts [43] improves the model capacity without increasing the training budget. Structural dropout [21, 17] exploits computationally efficient ways to regularize the model. Efficient attention [27, 10] reduces the quadratic time complexity for computing attention. Distributed training systems [39, 22] reduce training time by leveraging more computational resources. Our work on reducing numerical precision is orthogonal with these directions.

## 3 Forward Propagation

Neural network training is an iterative optimization procedure with stochastic gradients computed by forward and backpropagation. We accelerate forward and back propagation with 4-bit integer

(INT4) arithmetic. We first describe the forward propagation of our training procedure. The forward propagation can be formulated as a composition of linear and non-linear (GeLU, normalization, softmax, etc.) operators. In our training procedure, we accelerate all the linear operators with INT4 arithmetic and leave all the less-computationally-intensive non-linear operators in the 16-bit floating-point (FP16) format. All linear operations in transformers can be written in a matrix multiplication (MM) form. For ease of presentation, we consider the acceleration of the following simple matrix multiplication throughout this paper:

$$\mathbf{Z} = \mathbf{X}\mathbf{W}^\top, \text{ where } \mathbf{Z} \in \mathbb{R}^{N \times C}, \mathbf{X} \in \mathbb{R}^{N \times D} \text{and } \mathbf{W} \in \mathbb{R}^{C \times D}. \tag{1}$$

The most predominant use case of such MM is the fully-connected layer. Consider a transformer with an input shape of *(batch size $S$, sequence length $T$, dimensionality $D$)*. The fully-connected layer can be written as Eq. (1) where $\mathbf{X}$ is the activation for $N = ST$ tokens, and $\mathbf{W}$ is the weight matrix. For attention layers, batch matrix multiplications (BMMs) might be required. Our proposed techniques can be applied to BMMs, and we leave the discussion of BMMs in Appendix. A.1.

### 3.1 Learned Step Size Quantization

To accelerate training, the forward propagation must be computed with integer arithmetic. We leverage the *learned step size quantizer* (LSQ) [16] for this purpose. LSQ is a static quantization method whose quantization scale does not depend on the input, and is thus cheaper than dynamic quantization methods [23], which need to compute the quantization scale dynamically per iteration.

Given a FP matrix $\mathbf{X}$, LSQ *quantizes* $\mathbf{X}$ to integer with

$$\text{int}_{s_X}(\mathbf{X}) := \lfloor \text{clamp}(\mathbf{X}/s_X, -Q_N, Q_P) \rceil, \tag{2}$$

where $s_X$ is a learnable scalar parameter, clamp restricts its input to the range $[-Q_N, Q_P]$, $\lfloor \cdot \rceil$ is a rounding operation, and $\mathbf{X}/s_X$ is computed elementwise. The resultant matrix takes values from $\{-Q_N, -Q_N + 1, \ldots, Q_P\}$. Since we aim to perform INT4 MMs, we set $Q_N = Q_P = 7$. The integer matrix can be *dequantized* back to FP through $\text{float}(\text{int}_{s_X}(\mathbf{X})) = s_X \text{int}_{s_X}(\mathbf{X}) \approx \mathbf{X}$.

With LSQ, Eq. (1) can be computed approximately as $\mathbf{Y} = \mathbf{X}\mathbf{W}^\top \approx s_X s_W \text{int}_{s_X}(\mathbf{X}) \text{int}_{s_W}(\mathbf{W})^\top$, where the INT4 MM $\text{int}_{s_X}(\mathbf{X}) \text{int}_{s_W}(\mathbf{W})^\top$ can be implemented efficiently on hardware.

**Remark:** Quantization-aware training (QAT) [9, 63, 67, 23, 12, 11, 44, 60, 45, 49, 64, 2, 18, 55] is an *inference acceleration* technique which trains networks with quantizers inserted in the forward propagation graph, so the trained network can perform efficiently during inference. QAT can compress activation/weights to extremely low precision (e.g. 1-2 bits). It is tempting to think that directly applying a quantizer for QAT to FQT can lead to similar low activation/weights bit-width. However, even only quantizing the forward propagation for FQT is much more challenging than QAT because: (1) QAT requires a converged full-precision model as initialization [16] and/or as a teacher model for knowledge distillation [2]; (2) QAT can adopt expensive multi-stage training pipelines without worrying about the convergence speed [32], while FQT algorithm must converge as fast as full-precision training algorithms to be useful; (3) QAT may approximate the discrete quantizer with continuous functions during training [19], which cannot be implemented with integer arithmetic. Due to these challenges, it is still an open problem to do FQT with 4-bit activations/weights.

### 3.2 Activation Outliers

Simply applying LSQ for FQT with 4-bit activation/weights leads to accuracy degradation due to *activation outliers* [58]. As shown in Fig. 1(a), activations have some outlier entries, which are much larger in magnitude than other entries. In this case, the step size $s_X$ poses a trade-off between quantization granularity and representable numerical range. If $s_X$ is large, we can represent the outliers well at the expense of representing most other entries in a very coarse manner. On the other hand, if $s_X$ is small, we have to truncate the entries outside the range $[-Q_N s_X, Q_P s_X]$. Unfortunately, the transformers tend to store information in these outliers, and such truncation would seriously harm accuracy (see Sec. 5.2 for details). The outlier problem is particularly significant when the training task is to fine-tune a pre-trained model on some new downstream tasks, since the pre-train model contains more outliers [58] than random initialization.

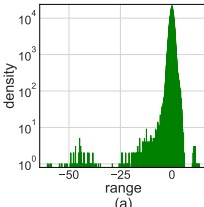 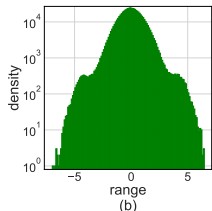 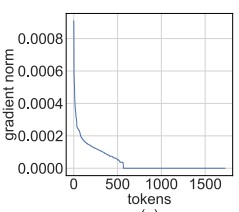 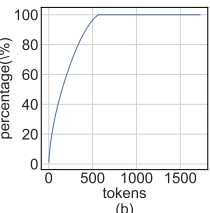

Figure 1: Histogram of activation of the `linear-1-2` layer in a BERT-base-uncased model. (a) Original activation distribution; (b) Hadamard-transformed activation distribution.

Figure 2: (a) The distribution of gradient norm along the token dimension. (b) The cumulative sum of the top X values as a percentage of the sum of all norms along the token dimension.

There exists some works to handle activation outliers for post-training quantization (PTQ). Outlier Suppression [56] discover that LayerNorms amplify outliers, and propose Gamma Migration and Token-Wise Clipping to solve this issue and achieves 6-bit BERT PTQ without too much degradation. SmoothQuant [58] migrates the quantization difficulty of activation outliers to weights and achieves 8-bit PTQ for large language models, such as OPT-175B. Outlier Channel Splitting [66] duplicates channels containing outliers with small overhead on the size of the network. However, these methods mainly focus on PTQ or QAT, and seldom successfully deal with ultra-low 4-bit training.

### 3.3 Hadamard Quantization

We propose a *Hadamard quantizer* (HQ) to solve the outlier problem. Its main idea is to quantize the matrices *in another linear space* which has fewer outliers.

The outliers in activation matrices form a feature-wise structure [58]. They are typically concentrated on a few dimensions, i.e., only a few columns of $\mathbf{X}$ are significantly larger than others. Hadamard transform [48] is a linear transformation, which can amortize the outliers into other entries. Specifically, the Hadamard transform $\mathbf{H}_k$ is a $2^k \times 2^k$ matrix, where

$$\mathbf{H}_0 = [1], \quad \mathbf{H}_k = \tfrac{1}{\sqrt{2}} \left[\mathbf{H}_{k-1} \quad \mathbf{H}_{k-1}; \mathbf{H}_{k-1} \quad -\mathbf{H}_{k-1}\right].$$

Hadamard matrices are orthogonal and symmetric: $\mathbf{H}_k = \mathbf{H}_k^\top = \mathbf{H}_k^{-1}$, so $\mathbf{H}_k\mathbf{H}_k = \mathbf{I}, \forall k \geq 0$. Consider any coordinate row vector* $\mathbf{e}_i^\top \in \mathbb{R}^{2^k}$, we have $\mathbf{e}_i^\top\mathbf{H}_k = 2^{-k/2}\mathbf{1}_{2^k}, \forall i$, where $\mathbf{1}_{2^k} = (\pm1, \pm1, \ldots, \pm1)$ is a $2^k$-dimensional vector with all of its values being 1 or $-1$. This demonstrates the extreme case when a single outlier dominates all the rest dimensions. In this case, Hadamard transformation effectively turns the vector into a quantization-friendly all-one-vector. The practical effect of the Hadamard transform on suppressing activation outliers is demonstrated in Fig. 1(b).

HQ uses a block-diagonal transformation matrix $\mathbf{H} \in \mathbb{R}^{D \times D}$: $\mathbf{H} = \text{BlockDiag}(\mathbf{H}_k, \ldots, \mathbf{H}_k)$, where $D$ is a multiple of $2^k$. To suppress outliers, we quantize a transformed version of $\mathbf{X}$ and $\mathbf{W}$:

$$\mathbf{X} = (\mathbf{XH})\mathbf{H}^\top \approx s_X \text{int}_{s_X} (\mathbf{XH}) \mathbf{H}^\top, \quad \mathbf{W} = (\mathbf{WH})\mathbf{H}^\top \approx s_W \text{int}_{s_W} (\mathbf{WH}) \mathbf{H}^\top.$$

Combining the quantized matrices, we get

$$\mathbf{Y} = \mathbf{XW}^\top \approx s_X s_W \text{int}_{s_X} (\mathbf{XH}) \mathbf{H}^\top \mathbf{H} \text{int}_{s_W} \left(\mathbf{H}^\top\mathbf{W}^\top\right) \quad = s_X s_W \text{int}_{s_X} (\mathbf{XH}) \text{int}_{s_W} \left(\mathbf{H}^\top\mathbf{W}^\top\right), \tag{3}$$

where the inverse transformations cancel with each other, and the MM can be implemented as:

> **Procedure** `HQ-MM`
> 1. Compute $\mathbf{XH}$ and $\mathbf{H}^\top\mathbf{W}^\top$ in FP16.
> 2. Quantize the resultant matrices to INT4 by LSQ.
> 3. Multiply the two INT4 matrices.
> 4. Dequantize the resultant INT32 matrix to FP16 by multiplying $s_X s_W$.

For time complexity, Step 1 takes $O(2^k N(D + C))$ FP16 multiply-accumulates (MACs); Step 2

---

*A vector which $i$-th dimension is 1, and all other dimensions are 0.

and Step 4 takes $O(N(D + C))$ FP16 MACs in total; and Step 3 takes $O(NDC)$ INT4 MACs. Comparing with the plain LSQ Eq. (2), the amount of FP16 MACs increases by $2^k$ times, from $O(N(D + C))$ to $O(2^k N(D + C))$. However, our HQ-MM is still much cheaper than an FP16 MM given $2^k \ll D$ and $2^k \ll C$. The number $k$ shows a tradeoff between the ability to suppress outliers and computation complexity. Larger $k$ allows for amortizing the outlier within a larger horizon, at the cost of being more expensive. We propose an adaptive algorithm to choose $k$ for each activation depending on the outlier scale, as discussed in Appendix A.5. The typical value is $k = 5$, while the dimensionality $C$ and $D$ ranges from 768 to 4096.

## 4   Backpropagation

We now consider accelerating the backpropagation of the linear layer with INT4 operations. The linear operator HQ-MM defined in Eq. (3) has four inputs: activation $\mathbf{X}$, weight $\mathbf{W}$, and step sizes $s_X$, $s_W$. Given the output gradient $\nabla_{\mathbf{Y}}\mathcal{L}$ w.r.t. some loss function $\mathcal{L}$, we need to compute the gradient of all four inputs. We discuss the computation of activation/weight gradients in this section, and left the discussion of step size gradients to Appendix A.3. For simplicity, we omit $\mathcal{L}$ and simply use $\nabla_{\mathbf{Y}}$ to denote the gradient in the following text.

By the straight-through estimator $\lfloor x \rceil' = 1$ [5] and the chain rule, we have

$$\nabla_{\mathbf{W}} = s_X \left( \nabla_{\mathbf{Y}}^{\top} \hat{\mathbf{X}} \circ \mathbb{I}_W \right) \mathbf{H}^{\top}, \quad \nabla_{\mathbf{X}} = s_W \mathbb{I}_X \circ \nabla_{\mathbf{Y}} \hat{\mathbf{W}} \mathbf{H}^{\top}, \tag{4}$$

where we define $\hat{\mathbf{X}} = \mathrm{int}_{s_X}(\mathbf{X}\mathbf{H})$, $\hat{\mathbf{W}} = \mathrm{int}_{s_W}(\mathbf{W}\mathbf{H})$, $\mathbb{I}_X = \mathbb{I}(-Q_N \leq \mathbf{X}/s_X \leq Q_P)$, and $\mathbb{I}_W = \mathbb{I}(-Q_N \leq \mathbf{W}/s_W \leq Q_P)$. For computing the gradients, three types of matrix multiplications are required:

1. The element-wise multiplication $\circ$ of a 0/1 matrix $\mathbb{I}_X$ (or $\mathbb{I}_W$) with another INT4 (or INT32) matrix. This operation has low time complexity.
2. The multiplication of an INT32 matrix with an FP16 block-wise Hadamard matrix $s_W \mathbf{H}^{\top}$, which also has low-time complexity, as discussed in Sec. 3.3.
3. The multiplication of the FP16 gradient $\nabla_{\mathbf{Y}}$ with an INT4 matrix $\hat{\mathbf{X}}$ or $\hat{\mathbf{W}}$, which we will accelerate by quantizing $\nabla_{\mathbf{Y}}$ to INT4.

In the rest of this section, we will discuss quantization methods to compute the "type 3" MMs $\nabla_{\mathbf{Y}}^{\top} \hat{\mathbf{X}}$ and $\nabla_{\mathbf{Y}} \hat{\mathbf{W}}$. We quantize $\nabla_{\mathbf{Y}}$ dynamically for each MM, while $\hat{\mathbf{X}}$ and $\hat{\mathbf{W}}$ have been already calculated in forward propagation in Section. 3. We start by discussing the structure of the gradient.

### 4.1   Structural Sparsity of Gradients

We note that the gradient matrix $\nabla_{\mathbf{Y}}$ tends to be very sparse along the training process. Furthermore, the sparsity has a structure: few rows (i.e., tokens) of $\nabla_{\mathbf{Y}}$ have large entries, while most other rows are close to an all-zero vector. We illustrate this by plotting the histogram of per-row norm $\|(\nabla_{\mathbf{Y}})_{i,:}\|$ for all the rows $i$ in Fig. 2.

Such a structural sparsity arises from the heavy overparameterization [62] of modern neural networks. During almost the entire training process, the network operates in the overparameterized scheme [34], where it can fit most training data well, except for a few hard examples. Therefore, the (activation) gradient will be close to zero for well-fitted data points. We find that for pretraining tasks, such structural sparsity quickly emerges after only a few training epochs. For fine-tuning tasks, the gradient is always sparse during the whole training process.

### 4.2   Bit Splitting and Leverage Score Sampling

Here, we discuss how to design gradient quantizers to accurately compute the MMs during backprop-agation by leveraging structural sparsity. The high-level idea is that many rows of the gradient are so small that they have little impact on the parameter gradient, yet they waste abundant computation. On the other hand, the large rows cannot be accurately represented with INT4. We drop some small rows and use the saved computation to represent large rows more accurately.

First, we propose *bit splitting* (BS), which splits a full-precision matrix as higher and lower 4 bits:

$$\nabla_{\mathbf{Y}} \approx s_\uparrow \nabla_{\mathbf{Y}}^\uparrow + s_\downarrow \nabla_{\mathbf{Y}}^\downarrow, \tag{5}$$

where $s_\uparrow, s_\downarrow$ are two floating-point scalars, and $\nabla_{\mathbf{Y}}^\uparrow, \nabla_{\mathbf{Y}}^\downarrow$ are INT4 matrices representing the higher and lower 4 bits, respectively. BS can be implemented by first quantizing $\nabla_{\mathbf{Y}}$ to INT4 as $\nabla_{\mathbf{Y}} \approx s_\uparrow \nabla_{\mathbf{Y}}^\uparrow$ and then quantize the residual to INT4 as $\nabla_{\mathbf{Y}} - s_\uparrow \nabla_{\mathbf{Y}}^\uparrow \approx s_\downarrow \nabla_{\mathbf{Y}}^\downarrow$. BS can be viewed as an INT8 representation of a matrix, where $\nabla_{\mathbf{Y}}^\uparrow$ and $\nabla_{\mathbf{Y}}^\downarrow$ are the higher and lower 4 bits of the INT8 representation. Next, we discuss how to compute the weight and activation gradient.

**Weight Gradient**  As discussed earlier, weight gradient involves the matrix multiplication $\nabla_{\mathbf{Y}}^\top \hat{\mathbf{X}}$, where $\nabla_{\mathbf{Y}} \in \mathbf{R}^{N \times C}$ and $\hat{\mathbf{X}}$ is an $N \times D$ INT4 matrix. By Eq. (5):

$$\nabla_{\mathbf{Y}}^\top \hat{\mathbf{X}} \approx (s_\uparrow \nabla_{\mathbf{Y}}^{\uparrow}{}^\top + s_\downarrow \nabla_{\mathbf{Y}}^{\downarrow}{}^\top) \hat{\mathbf{X}} = \nabla_{\mathbf{Y}}^{\updownarrow}{}^\top \mathbf{X}^\updownarrow, \tag{6}$$

where we define $\nabla_{\mathbf{Y}}^\updownarrow = [s_\uparrow \nabla_{\mathbf{Y}}^\uparrow; s_\downarrow \nabla_{\mathbf{Y}}^\downarrow]^\top \in \mathbb{R}^{2N \times C}$ and $\hat{\mathbf{X}}^\updownarrow = [\hat{\mathbf{X}}; \hat{\mathbf{X}}]$ to be a $2N \times D$ INT4 matrix. Eq. (6) represents the product of an INT8 $\nabla_{\mathbf{Y}}^\updownarrow$ and an INT4 $\hat{\mathbf{W}}$, and can be implemented by two INT4 MMs $\nabla_{\mathbf{Y}}^{\uparrow}{}^\top \hat{\mathbf{X}}$ and $\nabla_{\mathbf{Y}}^{\downarrow}{}^\top \hat{\mathbf{X}}$. Such MM is rather accurate since $\nabla_{\mathbf{Y}}$ is represented with 8 bits.

However, comparing to a naïve quantization of $\nabla_{\mathbf{Y}}$ to INT4, BS doubles the amount of INT4 operations for MM. We propose *leverage score sampling* (LSS) to cut the operations of Eq. (5) by half, to the same amount as the naïve MM $s_\uparrow \nabla_{\mathbf{Y}}^\uparrow \hat{\mathbf{X}}$. Noticing that the MM Eq. (6) can be written as the sum of $2N$ rank-1 matrices:

$$\nabla_{\mathbf{Y}}^{\updownarrow}{}^\top \mathbf{X}^\updownarrow = \sum_{i=1}^{2N} \nabla_{\mathbf{Y}_{:,i}}^{\updownarrow}{}^\top \mathbf{X}_i^\updownarrow = \sum_{i=1}^{2N} \nabla_{\mathbf{W}_i}, \tag{7}$$

where $\nabla_{\mathbf{W}_i} = \nabla_{\mathbf{Y}_{:,i}}^{\updownarrow}{}^\top \mathbf{X}_i^\updownarrow$. Due to the sparsity of $\nabla_{\mathbf{Y}}$, the matrices $\nabla_{\mathbf{W}_i}$ differ in magnitude and small matrices can be discarded without having a big influence on the result.

Our proposed LSS assigns each $\nabla_{\mathbf{W}_i}$ a probability $p_i \in [0,1], i = 1, \cdots, 2N$, that satisfies $\sum_{i=1}^{2N} p_i = N$. We define random masks $m_i \sim \text{Bern}(p_i)$ and mask matrix $\tilde{\mathbf{M}}$, and approximate it as

$$\nabla_{\mathbf{Y}}^{\updownarrow}{}^\top \mathbf{X}^\updownarrow \approx \nabla_{\mathbf{Y}}^{\updownarrow}{}^\top \tilde{\mathbf{M}} \mathbf{X}^\updownarrow = \sum_{i=1}^{2N} \frac{m_i}{p_i} \nabla_{\mathbf{Y}_{:,i}}^{\updownarrow}{}^\top \mathbf{X}_i^\updownarrow, \text{where } \tilde{\mathbf{M}} = \text{diag}\left(\frac{m_1}{p_1}, \ldots, \frac{m_{2N}}{p_{2N}}\right),$$

which is an unbiased approximation since $\mathbb{E}\left[\nabla_{\mathbf{Y}}^{\updownarrow}{}^\top \tilde{\mathbf{M}} \mathbf{X}^\updownarrow\right] = \nabla_{\mathbf{Y}}^{\updownarrow}{}^\top \mathbb{E}\left[\tilde{\mathbf{M}}\right] \mathbf{X}^\updownarrow = \nabla_{\mathbf{Y}}^{\updownarrow}{}^\top \mathbf{X}^\updownarrow$.

In expectation, there are only $N$ nonzero $m_i$s. Therefore, LSS reduces the cost of MM by half. For LSS to be accurate, we minimize its variance. We have:

**Proposition 4.1.** *(LSS variance for weight gradient)*

$$\text{Var}\left[\sum_{i=1}^{2N} \frac{m_i}{p_i} \nabla_{\mathbf{Y}_{:,i}}^{\updownarrow}{}^\top \mathbf{X}_i^\updownarrow\right] = \sum_{i=1}^{2N} \frac{1-p_i}{p_i} \|\nabla_{\mathbf{Y}_{i,:}}^\updownarrow\|^2 \|\mathbf{X}_{i,:}^\updownarrow\|^2, \text{where } \text{Var}[\mathbf{X}] := \mathbb{E}\left[\|\mathbf{X} - \mathbb{E}\mathbf{X}\|\right]_F^2.$$

The coefficient $c_i := \|\nabla_{\mathbf{Y}_{i,:}}^\updownarrow\| \|\mathbf{X}_{i,:}^\updownarrow\|$ is called the *leverage score*, which can be easily computed in low time complexity. When $p_i \propto c_i$, the variance attends its minimum due to Cauchy inequality:

$$\sum_{i=1}^{2N} \frac{1}{p_i} \|\nabla_{\mathbf{Y}_{i,:}}^\updownarrow\|^2 \|\mathbf{X}_{i,:}^\updownarrow\|^2 = \sum_{i=1}^{2N} \frac{c_i^2}{p_i} = \sum_{i=1}^{2N} \frac{c_i^2}{p_i} \sum_{i=1}^{2N} p_i \geq (\sum_{i=1}^{2N} c_i)^2,$$

where the equality holds when $p_i \propto c_i$. Intuitively, LSS can approximate the MM Eq. (7) well with significantly lower computational cost when the leverage scores $\{c_i\}$ are diverse, which is indeed the case as shown in Fig. 2.

Define $\mathbf{M}^\uparrow$ to be the top-left $N \times N$ submatrix of $\tilde{\mathbf{M}}$ and $\mathbf{M}^\downarrow$ to be the bottom-right one, we have

$$\nabla_{\mathbf{Y}}^{\updownarrow}{}^\top \tilde{\mathbf{M}} \mathbf{X}^\updownarrow = s_\uparrow \nabla_{\mathbf{Y}}^{\uparrow}{}^\top \tilde{\mathbf{M}}^\uparrow \hat{\mathbf{X}} + s_\downarrow \nabla_{\mathbf{Y}}^{\downarrow}{}^\top \tilde{\mathbf{M}}^\downarrow \hat{\mathbf{X}},$$

Table 1: Results on language model fine-tuning, transformer pretraining, and vision transformers fine-tuning and pretraining. Standard deviation is reported as subscript. FT refers to Fine-tuning, and PT refers to Pre-training. For WMT the result of 25.4 is result of Ultra-Low, not INT8.

| | | | | BASELINES | | 4-BIT TRAINING METHODS | |
|---|---|---|---|---|---|---|---|
| DATASET | TRAIN TYPE | MODEL | METRIC NAME | FP | INT8 | LSQ+LUQ | HQ+LSS |
| GLUE-dev | FT | BERT-BASE | AVG | $82.67_{0.24}$ | $81.45_{0.13}$ | $75.29_{0.52}$ | $\mathbf{80.81_{0.31}}$ |
| | | BERT-LARGE | AVG | $84.57_{0.42}$ | $82.74_{0.24}$ | $55.93_{2.47}$ | $\mathbf{82.25_{0.58}}$ |
| SQUAD v1 | FT | BERT-BASE | F1 | $88.32_{0.30}$ | $88.42_{0.20}$ | $85.75_{0.31}$ | $\mathbf{87.60_{0.25}}$ |
| SQUAD v2 | FT | BERT-BASE | F1 | $76.04_{0.68}$ | $75.63_{0.07}$ | $71.02_{0.41}$ | $\mathbf{74.63_{0.18}}$ |
| ADVERSARIAL QA | FT | BERT-BASE | F1 | $40.99_{0.38}$ | $40.17_{0.58}$ | $31.85_{0.30}$ | $\mathbf{38.70_{0.77}}$ |
| SWAG | FT | BERT-BASE | ACC | $79.84_{0.10}$ | $79.18_{0.19}$ | $70.79_{1.20}$ | $\mathbf{77.49_{0.16}}$ |
| CONLL | FT | BERT-BASE | ACC | $93.38_{0.08}$ | $93.13_{0.14}$ | $87.63_{0.39}$ | $\mathbf{91.90_{0.48}}$ |
| WMT | PT | TRANSFORMER-BASE | BLEU | 27.5 | 25.4(ULTRA LOW) | 27.17 | - |
| | | | SACREBLEU | 26.5 | - | - | 25.57 |
| CIFAR10 | FT | VIT-B/32 | TOP1 ACC | $98.77_{0.03}$ | $98.59_{0.02}$ | $97.76_{0.10}$ | $\mathbf{98.36_{0.05}}$ |
| | | VIT-L/32 | | 98.98 | 98.76 | 98.38 | **98.47** |
| CIFAR100 | FT | VIT-B/32 | TOP1 ACC | $91.94_{0.11}$ | $90.99_{0.07}$ | $88.63_{0.085}$ | $\mathbf{89.78_{0.06}}$ |
| | | VIT-L/32 | | 93.07 | 92.2 | 90.97 | **91.13** |
| IMAGENET1K | FT | VIT-B/32 | TOP1 ACC | 81.88 | 80.42 | 77.25 | **79.18** |
| | | VIT-L/32 | | 81.62 | 81.3 | 77.41 | **80.06** |
| | | VIT-L/16 | | 84.55 | 83.05 | 82.4 | **82.61** |
| | PT | DEIT-SMALL | TOP1 ACC | 73.1 | 70.95 | **69.96** | 69.18 |

which can be implemented by two INT4 MMs with sampled rows/columns. Putting everything together, we propose the following MM procedure to compute the weight gradient:

---
**Procedure** `LSS-MM`
1. Quantize $\nabla_{\mathbf{Y}}$ with BS to obtain $\nabla_{\mathbf{Y}}^{\uparrow}$ and $\nabla_{\mathbf{Y}}^{\downarrow}$ in INT4.
2. Compute the leverage score $\|\nabla_{\mathbf{Y}_{i,:}}^{\updownarrow}\|\|\mathbf{X}_{i,:}^{\updownarrow}\|$ in FP16.
3. Sample the masks $\{m_i\}$.
4. Sample rows of $\nabla_{\mathbf{Y}}$ and $\hat{\mathbf{X}}$ given the masks $\{m_i\}$.
5. Compute INT4 MMs ${\nabla_{\mathbf{Y}}^{\uparrow}}^{\top}\tilde{\mathbf{M}}^{\uparrow}\hat{\mathbf{X}}$ and ${\nabla_{\mathbf{Y}}^{\downarrow}}^{\top}\tilde{\mathbf{M}}^{\downarrow}\hat{\mathbf{X}}$,
6. Dequantize and sum up the resultant INT32 matrices to obtain the FP16 result ${\nabla_{\mathbf{Y}}^{\updownarrow}}^{\top}\tilde{\mathbf{M}}\mathbf{X}^{\updownarrow}$.

---

As $\tilde{\mathbf{M}}$ only has $N$ non-zero elements in expectation, the two matrix multiplications in Step 5 take about $2NCD$ INT4 MACs, which aligns with the cost of the naïve MM $s_{\uparrow}\nabla_{\mathbf{Y}}^{\uparrow}\hat{\mathbf{X}}$. The overhead of all the other steps is $O(NC + ND)$ in total.

**Activation Gradient** Similar to the previous discussion, the gradient of input can be written as

$$\nabla_{\mathbf{Y}}\hat{\mathbf{W}} \approx (s_{\uparrow}\nabla_{\mathbf{Y}}^{\uparrow} + s_{\downarrow}\nabla_{\mathbf{Y}}^{\downarrow})\hat{\mathbf{W}} = s_{\uparrow}\nabla_{\mathbf{Y}}^{\uparrow}\hat{\mathbf{W}} + s_{\downarrow}\nabla_{\mathbf{Y}}^{\downarrow}\hat{\mathbf{W}} = \left(\hat{\mathbf{I}}^{\updownarrow}\nabla_{\mathbf{Y}}^{\updownarrow}\right)\hat{\mathbf{W}}, \qquad (8)$$

where we define $\nabla_{\mathbf{Y}}^{\updownarrow} = [s_{\uparrow}\nabla_{\mathbf{Y}}^{\uparrow}; s_{\downarrow}\nabla_{\mathbf{Y}}^{\downarrow}] \in \mathbb{R}^{2N \times C}$ and $\hat{\mathbf{I}}^{\updownarrow} = [\mathbf{I} \quad \mathbf{I}]$ to be a $N \times 2N$ INT4 matrix, $\mathbf{I}$ is a $N \times N$ identity matrix. The original product can also be implemented by two INT4 MMs $\nabla_{\mathbf{Y}}^{\uparrow}\hat{\mathbf{W}}$ and $\nabla_{\mathbf{Y}}^{\downarrow}\hat{\mathbf{W}}$. But different from weight gradients, we now focus on $\hat{\mathbf{I}}^{\updownarrow}\nabla_{\mathbf{Y}}^{\updownarrow}$ in Eq. (8) and do leverage score sampling on this MM. A detailed discussion can be found in Appendix B.2, and we only present the leverage score here. Similarly, we write the MM as the sum of $2N$ smaller multiplications:

$$\hat{\mathbf{I}}^{\updownarrow}\nabla_{\mathbf{Y}}^{\updownarrow} = \sum_{i=1}^{2N} \hat{\mathbf{I}}_{:,i}^{\updownarrow}\nabla_{\mathbf{Y}i}^{\updownarrow} \approx \frac{m_i}{p_i}\sum_{i=1}^{2N}\nabla_{\mathbf{Y}_i},$$

where we define $\nabla_{\mathbf{Y}_i} = \hat{\mathbf{I}}_{:,i}^{\updownarrow}\nabla_{\mathbf{Y}i}^{\updownarrow}$ and associate the probability $p_i$ and Bernoulli mask $m_i \sim \text{Bern}(p_i)$ with the $i$ multiplication. The leverage score for activation gradient is $c_i := \|\nabla_{\mathbf{Y}i}^{\updownarrow}\|$, and the variance attains minimum when $p_i \propto c_i$. More details about the algorithm can be found at Appendix. A.3 On the implementation side, once the mask $\{m_i\}$ is known, we can decompose the MM Eq. (8) as two INT4 MMs: $\left(\hat{\mathbf{I}}^{\updownarrow}\tilde{\mathbf{M}}\nabla_{\mathbf{Y}}^{\updownarrow}\right)\hat{\mathbf{W}} = s_{\uparrow}\tilde{\mathbf{M}}^{\uparrow}\nabla_{\mathbf{Y}}^{\uparrow}\hat{\mathbf{W}} + s_{\downarrow}\tilde{\mathbf{M}}^{\downarrow}\nabla_{\mathbf{Y}}^{\downarrow}\hat{\mathbf{W}}$.

## 5 Experiments

We evaluate our INT4 training algorithm on a wide variety of tasks including language model fine-tuning, machine translation, and image classification. We implement our proposed `HQ-MM` and

`LSS-MM` algorithms with CUDA and cutlass[†], and the implementation details can be found in Appendix A. We replace all the floating-point linear operators with our INT4 implementation except simply using LSQ for embedding layers, and leaving the last classifier layer in full precision. We adopt default architectures, optimizers, schedulers, and hyper-parameters for all the evaluated models.

## 5.1 Converged Model Accuracy

We compare the accuracy of the converged model on various tasks in Table 1. The compared methods include full-precision training (FP), INT8 training [3](INT8), FP4 training [47] ("Ultra-low"), 4-bit logarithm quantization [8] with LSQ for activations and weights (LSQ+LUQ), and our algorithm which utilizes HQ for forward and LSS for backpropagation (HQ+LSS). Ultra-low does not have a public implementation, so we only report its performance from its original paper on the machine translation task. Except for the large machine translation task and the task of large vision transformers, we repeat each run by three times and report the standard deviation as subscripts in tables. We do not include any kind of knowledge distillation or data augmentation.

**Language model fine-tuning:** We use the pretrained BERT-base-uncased and BERT-large-uncased [24] model, and evaluate the performance of our method on GLUE dev-set [53], SQUAD [41], SQUADv2 [40], Adversarial QA [4], CoNLL-2003 [42] and SWAG [61] datasets. We present the average result of the bert-base-uncased and bert-large-uncased models on the GLUE dataset. The full results are listed in Appendix C.2. Compared with LSQ+LUQ, our method achieves $5.5\%$ improvement of accuracy on average for the bert-base model and achieves $> 25\%$ improvement of accuracy on average for the bert-large model. We further show the result on the SQUAD, SQUAD 2.0, Adversarial QA, CoNLL-2003, and SWAG datasets. On all of the tasks, compared with LSQ+LUQ, our method achieves better performance. We improve by $1.8\%$ and $3.6\%$ on SQUAD and SQUAD 2.0 compared to LSQ+LUQ, respectively. On the more difficult Adversarial QA, we improve by $6.8\%$ on F1 score. On SWAG we improve by $6.7\%$ and on CoNLL-2003 we improve by $4.2\%$ accuracy.

**Machine translation:** We also apply our method for pretraining. We train a Transformer-base [52] model on WMT 14 En-De dataset [6] for machine translation. Note that we reproduce this experiment with Fairseq's recipe [‡], which reports the SacreBleu score (26.5 for FP) [37], while Ultra-low and LUQ report the more optimistic original BLEU score (27.5 for FP) [36]. Our HQ+LSS has about $1.0\%$ BLEU degradation, which is smaller than $2.1\%$ of Ultra-low and higher than $0.3\%$ reported in the LUQ paper. Nevertheless, HQ+LSS still performs comparably with existing methods for this pretraining task, and it supports contemporary hardware.

**Image Classification:** We load ViT checkpoints pretrained on ImageNet21k [13], and fine-tune it on CIFAR-10, CIFAR-100 [28], and ImageNet1k. We use ViT-B/32 and ViT-L/32 for CIFAR datasets and use ViT-B/32, ViT-L/32 and ViT-L/16 for ImageNet1k. On CIFAR10 we achieve $< 0.5\%$ accuracy degradation, while LSQ+LUQ has $1\%$ degradation for ViT-B/32 and $0.6\%$ degradation for ViT-L/32. On CIFAR100, INT8 already has $\sim 1\%$ accuracy degradation, which shows its difficulty. We improve by $1.1\%$ accuracy for ViT-B/32 and $0.2\%$ accuracy for ViT-L/32 compared with LSQ+LUQ. On ImageNet1k, we improve by $2\%$ accuracy for ViT-B/32, $2.6\%$ accuracy for ViT-L/32 and $0.2\%$ for ViT-L/32 compared with LSQ+LUQ. We further test the effectiveness of our algorithm for pretraining a DeiT-Small model [51] on ImageNet1K, where HQ+LSS can still converge to similar accuracy level compared to LSQ+LUQ, while being more hardware friendly.

## 5.2 Ablation Study

Here, we conduct ablation studies to show the effectiveness of our forward and backward methods independently on the challenging CoLA dataset. To study the effectiveness of different quantizers for forward propagation, we leave backpropagation in FP16. The result is shown in Fig. 3(a). We first validate the claim in Sec. 3.2 that outliers are the main cause of accuracy degradation in quantized forward propagation. We test an "outlier" method which maintains $1\%$ largest activation entries in FP. The "outlier" method achieves good performance, which proves that outliers are indeed the most

---

[†]`https://github.com/NVIDIA/cutlass`
[‡]`https://github.com/facebookresearch/fairseq`

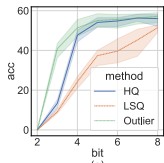 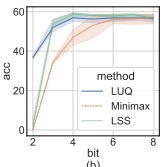 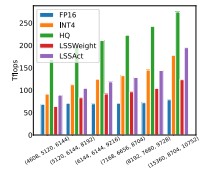 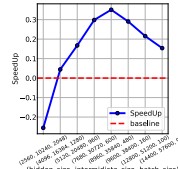 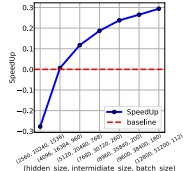

Figure 3: CoLA performance under different methods using different bits. (a) Comparison of forward methods. (b) Comparison of backward methods.

Figure 4: Comparison of basic FP16 MM, HQ, and LSS operators.

Figure 5: SpeedUp of our INT4 training algorithm compared with FP16 PyTorch AMP on (a) Bert-Large (b) Gpt2-base.

significant challenge of the transformer's forward quantization. The hardware-unfriendly "outlier" method serves as an upper bound of methods to handle outliers. Our HQ outperforms LSQ by better handling the outliers and achieves comparable results to maintaining the outliers.

We also investigated whether more granular quantizers, such as per-token quantization or per-channel quantization could be used to quantify outliers, or whether existing methods like SmoothQuant [58] could be used for INT4 FQT. The results are listed in Appendix C.3, and we find that without HQ, none of these methods achieve good accuracy under 4-bit quantization, and the result of HQ is not strongly affected when more granular quantization methods are applied.

For backpropagation, we compare a simple minimax quantizer [3], LUQ [8] and our LSS, and leave forward propagation in FP16. The minimax quantizer divides the numerical range from the minimum to the maximum into equally large quantization bins. The result is shown in Fig. 3(b). While the bit-width is higher than 2, our LSS achieves results that are comparable and even slightly higher than LUQ. Meanwhile, LSS is more hardware friendly as it requires only INT4 arithmetic.

## 5.3 Computational and Memory Efficiency

Finally, we demonstrate the potential of our method to accelerate neural network training by evaluating our prototypical implementation discussed in Appendix A.6. We emphasize that our implementation is not fully optimized. For example, the backward computation requires an INT4 MM in the form of $\mathbf{Y} = \mathbf{A}\mathbf{B}$, while cutlass only supports $\mathbf{Y} = \mathbf{A}\mathbf{B}^\top$, so explicit transpose is required. We also do not fuse the linear operators with nonlinearities and normalizations. Therefore, the results cannot fully reflect the potential of INT4 training algorithms. A fully optimized implementation requires heavy engineering, which exceeds the scope of our paper.

**Operator Speed:** We compare the throughput of our proposed `HQ-MM` (HQ), LSS for computing weight gradient (LSSWeight), LSS for computing activation gradient (LSSAct), and their average throughput (INT4) with a baseline tensor-core FP16 GEMM implementation (FP16) provided by cutlass in Fig. 4 on an Nvidia RTX 3090 GPU which has a peak throughput at 142 FP16 TFLOPs and 568 INT4 TFLOPs. As the matrix size grows, the overhead of quantization diminishes and our INT4 operators can be up to 2.2 times faster compared with FP16 MM. We further analyze the quantization overhead for each operator in Appendix C.5.

**Training Throughput:** We compare the training throughput of the FP16 PyTorch AMP and our INT4 training algorithm for training BERT [24] and GPT [38]-style language models on a system of 8 Nvidia A100 GPUs. We vary the hidden layer size, intermediate fully-connected layer size, and batch size, and plot the speedup of INT4 training in Fig. 5. Our INT4 training algorithm can achieve up to 35.1% speedup and an average of 15.8 % for BERT-style models and up to 26.5% speedup and an average of 19.2 % for GPT-style models. The training time can be found in Appendix C.4.

**Inference Speed:** We compare the inference speed of our algorithm with I-BERT [25] by comparing the speedup numbers reported in its original paper. Following the I-BERT paper, we compare the speedup of integer-based inference algorithms relative to a FP32 baseline on an Nvidia T4 GPU on the BERT-base and BERT-large models, and test with sequence lengths of 128 and 256. While I-BERT only reported speedup numbers for batch sizes 1, 2, 4, 8, we test the speedup for batch sizes

Table 2: The inference speed up compared with I-BERT. Since the open-sourced versions of I-BERT are based on fake quantization, we compare the inference speed of our algorithm with I-BERT by comparing the speedup numbers reported in its original paper with respect to FP32.

| SL | MODEL | METHOD | BATCH SIZE | | | | | | | | | | |
|---|---|---|---|---|---|---|---|---|---|---|---|---|---|
| | | | 1 | 2 | 4 | 8 | 16 | 32 | 64 | 128 | 256 | 512 | 1024 |
| 128 | BERT-BASE | I-BERT | 2.42 | 3.36 | 3.39 | 3.31 | - | - | - | - | - | - | - |
| | | OURS | 0.17 | 0.25 | 0.46 | 0.87 | 1.22 | 2.07 | **3.61** | **3.57** | **4.5** | **4.54** | **4.92** |
| | BERT-LARGE | I-BERT | 3.2 | 4 | 3.98 | 3.81 | - | - | - | - | - | - | - |
| | | OURS | 0.22 | 0.43 | 0.81 | 1.46 | 2.13 | 3.34 | **4.28** | **4.81** | **5.4** | **6.08** | **6.48** |
| 256 | BERT-BASE | I-BERT | 3.11 | 2.96 | 2.94 | 3.15 | - | - | - | - | - | - | - |
| | | OURS | 0.25 | 0.54 | 0.95 | 1.53 | 1.94 | **3.2** | **3.76** | **4** | **4.15** | **4.15** | **4.14** |
| | BERT-LARGE | I-BERT | 3.19 | 3.51 | 3.37 | 3.4 | - | - | - | - | - | - | - |
| | | OURS | 0.45 | 0.86 | 1.47 | 2.44 | 2.67 | **3.99** | **4.87** | **5.24** | **5.11** | **5.41** | OOM |

ranging from 1 to 1024 to better reflect the performance of throughput-oriented scenarios (such as a cloud language model service provider).

In Table 2 we report the speed up result. While I-BERT's speedup numbers seems to be insensitive to the batch size and sequence length, our speedup increases with the batch size. I-BERT shows up to 3.98x speedup for smaller batch size, while our algorithm can achieve higher speedup for batch sizes higher than 64, and eventually gives a speedup of 6.48x for a sequence length of 128 and a batch size of 1024.

For the BERT-base model, our method shows an inference speed improvement of 3.57-4.92 times compared to FP32 for batch sizes larger than 64. In comparison, I-BERT achieved an inference speed improvement of 2.42-3.39 times compared to FP32 when the batch size was small. For BERT-large, our method shows an inference speed improvement of 4.81-6.48 times compared to FP32 for batch sizes larger than 64. In comparison, I-BERT achieved an inference speed improvement of 3.20-4.00 times compared to FP32 when the batch size is small. Therefore, our algorithm can potentially achieve higher throughput than I-BERT.

## 6 Conclusions

We propose a hardware-friendly INT4 training method for transformers. By analyzing the properties of MMs in transformers, we propose HQ and LSS methods to quantize activations and gradients while preserving accuracy. On several important tasks, our method performs comparably or better than existing INT4 methods. Our work can be potentially extended beyond transformers to other MM-only architectures, such as MLP-Mixer [50], graph neural networks [26], and recurrent neural networks [20]. We leave it as a future direction.

**Broader Impacts:** Our algorithm can improve efficiency and reduce the energy consumption of training neural networks, which helps reduce the carbon footprint caused by deep learning. However, our efficient training algorithm might also facilitate the development of large language models with safety concerns for human beings; and malicious AI applications such as fake content generation.

**Limitations:** Our INT4 training algorithm does not support convolutional layers. Moreover, our algorithm cannot yet work well for those extremely large models such as OPT-175B. LSS utilizes gradient sparsity, and may not work well for certainty pretraining tasks. Finally, our algorithm may incur more memory accesses than FP16 training algorithms, which might affect the speedup.

## Acknowledgements

The authors would like to thank Weilin Zhao, Sixuan Ma, Xiaoxuan Liu, Ziteng Wang, Bingrui Li, Cheng Lu, and Zhiyuan Liu for valuable discussions and help on the training of large-scale language models. This work was supported by the National Key Research and Development Program of China (No. 2021ZD0110502), NSFC Projects (Nos. 62061136001, 62106123, 62076147, U19A2081, 61972224, 62106120), Tsinghua Institute for Guo Qiang, and the High Performance Computing Center, Tsinghua University. J.Z is also supported by the XPlorer Prize.

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

# A  Implementation Details

In this section, we present some works that need to be done to actually accelerate the training process on hardware.

## A.1  BMM in Attention

In attention, there are batch matrix multiplications (BMMs) that need to be dealt with. We now show that our method for MMs can be extended to BMMs.

Consider the following BMM product:

$$\mathbf{T} = \text{BMM}(\mathbf{Q}, \mathbf{K}^{\top}),$$

where we define $\mathbf{T} \in \mathbb{R}^{B \times N \times P}, \mathbf{Q} \in \mathbb{R}^{B \times N \times M}, \mathbf{K} \in \mathbb{R}^{B \times P \times M}$. The Hadamard matrix is defined as :

$$\hat{\mathbf{H}} = \text{Repeat}_B(\mathbf{H}) = \text{Repeat}_B(\text{BlockDiag}(\mathbf{H}_k, \ldots, \mathbf{H}_k)),$$

where $\hat{\mathbf{H}} \in \mathbb{R}^{B \times M \times M}, \mathbf{H} \in \mathbb{R}^{M \times M}, \mathbf{H}_k \in \mathbb{R}^{2^k \times 2^k}$. In this case,

$$\mathbf{T} \approx \text{BMM}\big(\text{BMM}(\mathbf{Q}, \hat{\mathbf{H}}), \text{BMM}(\mathbf{K}, \hat{\mathbf{H}})^{\top}\big),$$

which verifies that our HQ can be applied to BMMs.

For backward, the gradient of weight and activation can be calculated by the straight-through estimator $\lfloor x \rceil' = 1$ and the chain rule:

$$\nabla_{\mathbf{Q}} = s_Q \left(\text{BMM}(\nabla_{\mathbf{T}}^{\top}, \hat{\mathbf{K}}) \circ \mathbb{I}_Q\right) \mathbf{H}^{\top},$$

$$\nabla_{\mathbf{K}} = s_K \mathbb{I}_K \circ \text{BMM}(\nabla_{\mathbf{T}}, \hat{\mathbf{Q}}) \mathbf{H}^{\top} = s_K \text{BMM}(\mathbb{I}_K \circ \nabla_{\mathbf{T}}, \hat{\mathbf{Q}}) \mathbf{H}^{\top},$$

where we define $s_Q \in \mathbb{R}^B, s_k \in \mathbb{R}^B$ being the batch step size, $\hat{\mathbf{K}} = \text{int}_{s_K}\left(\text{BMM}(\mathbf{K}, \hat{\mathbf{H}})\right)$, $\hat{\mathbf{Q}} = \text{int}_{s_Q}\left(\text{BMM}(\mathbf{Q}, \hat{\mathbf{H}})\right)$, $\mathbb{I}_Q = \mathbb{I}(-Q_N \le \mathbf{Q}/s_Q \le Q_P)$, and $\mathbb{I}_K = \mathbb{I}(-Q_N \le \mathbf{K}/s_K \le Q_P)$.

Similar to Sec. 4.2, we only focus on $\text{BMM}(\nabla_{\mathbf{T}}^{\top}, \hat{\mathbf{K}})$ and $\nabla_{\mathbf{T}}$, since we do leverage sampling on them.

For $\text{BMM}(\nabla_{\mathbf{T}}^{\top}, \hat{\mathbf{K}})$, we define the sample probability $p_i$ and sample the $\tilde{\mathbf{M}}$ in the same way as MMs. The matrix can be computed as $\text{BMM}(\text{BMM}(\nabla_{\mathbf{T}}^{\updownarrow\top}, \hat{\tilde{\mathbf{H}}}), \hat{\mathbf{K}}^{\updownarrow})$, where $\hat{\tilde{\mathbf{H}}}$ is defined as $\text{CONCAT}(\tilde{\mathbf{H}}_1, \cdots, \tilde{\mathbf{H}}_B), \nabla_{\mathbf{T}}^{\updownarrow\top}$ and $\hat{\mathbf{K}}^{\updownarrow}$ follows the same definition of Eq. 6and the leverage score is $c_{b,i} := \|\nabla_{\mathbf{T}b,i,:}^{\updownarrow}\|\|\mathbf{K}_{b,i,:}^{\updownarrow}\|$ for $0 \le b \le B, 0 \le i \le 2M$.

For $\nabla_{\mathbf{T}}$, similarly, can be viewed as $\nabla_{\mathbf{T}} = \text{BMM}(\hat{\mathbf{I}}^{\updownarrow}, \nabla_{\mathbf{T}}^{\updownarrow})$, where we define $\nabla_{\mathbf{Y}}^{\updownarrow} = \text{CONCAT}([s_{\uparrow b}\nabla_{\mathbf{T}b}^{\uparrow}; s_{\downarrow b}\nabla_{\mathbf{T}b}^{\downarrow}]) \in \mathbb{R}^{B \times 2N \times P}$, $\hat{\mathbf{I}}^{\updownarrow} = \text{CONCAT}([\mathbf{I} \quad \mathbf{I}]) \in \mathbb{R}^{B \times N \times 2N}$, $s_{\uparrow b}, \nabla_{\mathbf{T}b}^{\uparrow}, s_{\downarrow b}, \nabla_{\mathbf{T}b}^{\downarrow}$ follows the definition of Eq.5. So it can be computed as $\text{BMM}(\text{BMM}(\hat{\mathbf{I}}^{\updownarrow}, \hat{\tilde{\mathbf{H}}}), \nabla_{\mathbf{T}}^{\updownarrow})$, where $\hat{\tilde{\mathbf{H}}}$ is defined as $\text{CONCAT}(\tilde{\mathbf{H}}_1, \cdots, \tilde{\mathbf{H}}_B)$, and the leverage score is $c_{b,i} := \|\nabla_{\mathbf{T},i,:}^{\updownarrow}\|$ for $0 \le b \le B, 0 \le i \le 2M$, which verifies that our LSS can be applied to BMM.

## A.2  Computing Leverage Score

In the previous discussion, we find the optimal sample probability $p_i$ that can minimize the variance of the gradient. However, it is likely for the proportional $p_i$ is larger than one, which is invalid for the Bernoulli distribution. Accordingly, we propose an algorithm to solve this issue.

Define the probability array as

$$P = [p_1^0, \cdots, p_{2N}^0], \sum_{i=1}^{2N} p_i^0 = N,$$

we first clamp the array to $p_i^1 \in [0, 1]$. In this case, $\sum_{i=1}^{2N} p_i^1 \leq N$, so we scale the $p_i$ which is smaller than 1 to make sure their sum is again $N$. However, this will probably introduce some more elements larger than 1, so we cycle through the above operations until all the $p_i \in [0, 1]$. This process will certainly stop, since if after the scaling operation, no element is larger than 1, then we get a valid distribution. Otherwise, the number larger than 1 is reduced by at least one, thus the process will halt after at most $O(N)$ times.

### A.3 Learning Quantizer Parameters

In this section, we discuss the detail of how to calculate the gradient of activation and quantization step size.

For gradient of activation, the coefficient $c_i := \|\nabla_{\mathbf{Y}i}^{\updownarrow}\|$ is the *leverage score* for activation gradient, and the variance achieves its minimum When $p_i \propto c_i$ by the Cauchy Inequality.

Putting everything together, we propose the following MM procedure to compute activation gradient:

---
**Procedure** `LSS-MM`
1. Quantize $\nabla_{\mathbf{Y}}$ with BS to obtain $\nabla_{\mathbf{Y}}^{\uparrow}$ and $\nabla_{\mathbf{Y}}^{\downarrow}$ in INT4.
2. Compute the leverage score $\|\nabla_{\mathbf{Y}i}^{\updownarrow}\|$ in FP16.
3. Sample the masks $\{m_i\}$.
4. Sample rows of $\nabla_{\mathbf{Y}}$ given the masks $\{m_i\}$.
5. Compute $\tilde{\mathbf{I}\mathbf{M}}^{\uparrow}\nabla_{\mathbf{Y}}^{\uparrow}$ and $\tilde{\mathbf{I}\mathbf{M}}^{\downarrow}\nabla_{\mathbf{Y}}^{\downarrow}$ by discard some of its rows.
6. Compute INT4 MMs $\tilde{\mathbf{I}\mathbf{M}}^{\uparrow}\nabla_{\mathbf{Y}}^{\uparrow}\hat{\mathbf{W}}$ and $\tilde{\mathbf{I}\mathbf{M}}^{\downarrow}\nabla_{\mathbf{Y}}^{\downarrow}\hat{\mathbf{W}}$.
7. Dequantize and sum up the resultant INT32 matrices to obtain the FP16 result $\hat{\mathbf{I}}^{\updownarrow}\nabla_{\mathbf{Y}}^{\updownarrow}\hat{\mathbf{W}}$.

---

The two matrix multiplications in Step 5 take about $2NCD$ INT4 MACs in expectation.

For the quantization step sizes. Following the chain rule, we have

$$\nabla_{s_W} = g(s_W)\nabla_{\mathbf{Y}}^{\top}\hat{\mathbf{X}} \circ \delta_{\mathbf{W}}(s_W), \ \nabla_{s_X} = g(s_X)\nabla_{\mathbf{Y}}\hat{\mathbf{W}} \circ \delta_{\mathbf{X}}(s_X),$$

where we define $g(s_W) = 1/\sqrt{Q_p N_W}$, $g(s_X) = 1/\sqrt{Q_p N_X}$, $N_W$ and $N_X$ being the number of elements of weight and activation, $\delta_{\mathbf{X}}(s_X) = \text{int}_{s_X}(\mathbf{X}) - \mathbb{I}_X \circ (\mathbf{X}/s_X)$, and $\delta_{\mathbf{W}}(s_W) = \text{int}_{s_W}(\mathbf{W}) - \mathbb{I}_W \circ (\mathbf{W}/s_W)$.

Notice that for computing $\nabla_{s_W}$ and $\nabla_{s_X}$, the most expensive MMs are $\nabla_{\mathbf{Y}}^{\top}\hat{\mathbf{X}}$ and $\nabla_{\mathbf{Y}}\hat{\mathbf{W}}$, which are already calculated through Eq. (7) and Eq. (8) during previous calculations, so it does not require extra computation. The elementwise multiplication with $\delta_{\mathbf{X}}(s_X)$ and $\delta_{\mathbf{W}}(s_W)$ requires minor computation.

### A.4 Cold Start Problem

There is a *cold start problem*. When the model is trained from scratch (i.e., from a random initialization), distributions of weights and activations can change rapidly in the early stage of optimization. In this case, jointly optimizing the quantization step size and the weights would cause the training to be unstable. As a remedy, we do not learn the step size in the first few iterations, and use a heuristic rule to dynamically set the step size for each tensor $\mathbf{X}$ to $2\text{mean}(\mathbf{X})/\sqrt{Q_p}$ in each iteration. For the WMT experiment, we stop to re-initialize at the second epoch, since the BLEU score is relatively high at the second epoch (about 21, while the final BLEU score after 30 epochs is 25.5). However, for the deit-small pretraining experiment, we re-initialize it at the end of the 10th epoch (we train it for 90 epochs in total), since at the end of the second epoch, the accuracy is not high, but at the end of the 10th epoch, the accuracy is relatively high.

### A.5 Choose hadamard matrix size

For the hadamard matrix, let the hadamard matrix to be $\mathbf{H} \in \mathbb{R}^{D \times D}$: $\mathbf{H} = \text{BlockDiag}(\mathbf{H}_k, \ldots, \mathbf{H}_k)$, where $D$ is a multiple of $2^k$. We first define

$$\bar{\mathbf{X}}_k = s_X \text{int}_{s_X}(\mathbf{X}\mathbf{H})\mathbf{H}^{\top}, \quad \bar{\mathbf{W}} = s_W \text{int}_{s_W}(\mathbf{W}\mathbf{H})\mathbf{H}^{\top},$$

where $\bar{\mathbf{X}}$ and $\bar{\mathbf{W}}$ can be viewed as an approximation of $\mathbf{X}$ and $\mathbf{W}$. Then, we define the quantization error to be $\mathrm{MSE}(\bar{\mathbf{X}}, \mathbf{X}) \times \mathrm{MSE}(\bar{\mathbf{W}}, \mathbf{W})$. We search for the optimal $k$ that can minimize this quantization error. For fine-tuning tasks, once the hadamard matrix size has been calculated, we fix it through the training process. For the pre-training task, since the distribution shifts greatly as we train the model, we empirically define a time when we re-initialize the hadamard matrix size and the LSQ step size. Usually, we do this when the first 2 epochs finish.

## A.6  GPU Implementation

In the previous discussion, we get to know `HQ-MM` and `LSS-MM` from an algorithm level, nevertheless it is not enough to actually implement it on hardware. In this section, we will delve deeper into hardware implementation details as well as extra limitations.

`HQ-MM` can be divided into 5 parts: Hadamard matrix multiplication, Quantize, Data Pack, INT4 GEMM, and Dequantize.

For the Hadamard matrix multiplication process, since it can be interpreted as a half float matrix multiplication process where the two matrices involved in the operation are input/weight matrix and hadamard matrix, respectively, we implement it in Python, because PyTorch MM uses CublassGemm and is more efficient then CutlassGemm.

In the quantize process, we quantize input/weight into INT4 data respectively, and also preserve a corresponding FP16 version for the LSQ Back Propagation process to use.

In the previous discussion, we assume the quantize part of `HQ-MM` is quantizing the resultant matrices to INT4, however, the smallest representation unit of data is INT8. As a result, we actually use INT8 data type to represent quantized data and pack two adjacent data into one data using $(data[1] << 4)|(data[0]\&15)$ in the data packing process, which means we use one INT8 data to represent two adjacent INT4 data. With both input matrices' data packed in this way, we then use cutlass tensor-core INT4 GEMM to do the matrix multiplication.

For the GEMM process, we choose Nvidia CutlassGemm because it's the most efficient open-source operator library we can find. We use INT4 Tensor Core Gemm for our implementation and it requires the two input matrices A&B to be RowMajor and ColMajor, respectively. Since the default Pytorch tensor is RowMajor, we have to use Transpose+Contiguous operations to make it ColMajor, which is very time-consuming and needs further optimization in the future.

Finally, we dequantize the INT GEMM result back into FP16 output using a dequantize kernel, which is the final output of the forward kernel.

As compared, `LSS-MM` is more complicated, and can be divided into 7 parts: Quantization of higher lower 4-bit, Leverage Score Calculating, Sampling, Data Pack, INT4 GEMM, Dequantize, and LSQ Back Propagation.

In the Quantize process, we fuse the quantize operation of higher 4-bit and lower 4-bit into a single kernel for acceleration. In the Leverage Score Calculating process, we use the quantized INT8 data to calculate the score and scale up it in the final because integer arithmetic is far more efficient than float arithmetic.

In the sampling process, we sample out rows/columns given the previously calculated leverage score. Note that in Section. A.2, we repeat our proposed algorithm for several loops to sample out specific elements, which is effective but not efficient. According to experiments, however, we notice that simply selecting elements whose leverage score is bigger than 0 can also work well, even better than our proposed algorithm in some cases. So in real quantization implementation, we just sample out rows/ columns whose Euclidean norm is bigger than 0 to accelerate our training process.

Pack, Gemm, and Dequantize processes are as similar as before. It's worth noting that for Int4 Tensor Core Gemm, suppose two input matrices have shape $M \times K$ and $K \times N$, $K$ needs to be a multiple of 32 so that the Tensor core Gemm address can be aligned. We do not need to consider this in the Forward Propagation process because the input data shape always satisfies. However, in the Back Propagation process, the matrix shape may not meet the requirement after sampling. As a result, we need zero_padding the sampled matrix so that $K$ can be a multiple of 32.

Finally, we utilize the dequantized data to do the LSQ Back Propagation. We also fuse all operations into a single Cuda kernel for acceleration, and the metric remains.

Besides the component of `HQ-MM` and `LSS-MM`, there is still something that needs to be mentioned.

1. We omit the Quantization and Leverage Score Calculating process in LSSinput, and use the same value as LSSWeight to accelerate the training process.

2. For Element-Wise kernel, we set block size as 256, grid size as input.numel()/256. For Reduction kernels like sum and min/max, we set block size as 32, grid size as RowNum, reducing elements in each row to the first 32 elements. We find this setting to be most efficient through experiments.

## B  Proofs.

In this section, we present the proofs of the leverage score.

### B.1  Proof of Proposition. 4.1

**Proposition B.1.** *(LSS variance for weight gradient)*

$$\mathrm{Var}\left[\sum_{i=1}^{2N}\frac{m_i}{p_i}\nabla_{\mathbf{Y}_{:,i}}^{\updownarrow}{}^{\top}\mathbf{X}_i^{\updownarrow}\right]=\sum_{i=1}^{2N}\frac{1-p_i}{p_i}\|\nabla_{\mathbf{Y}_{i,:}}^{\updownarrow}\|^2\|\mathbf{X}_{i,:}^{\updownarrow}\|^2.$$

*Proof.*

$$
\begin{aligned}
Var(\nabla_{\mathbf{W}}) &= Var\Big(\sum_{i=1}^{2N}\frac{1}{p_i}(m_i\nabla_{\mathbf{Z}_{:,i}}^{\updownarrow}{}^{\top}\mathbf{X}_i^{\updownarrow})\Big)\\
&= Var\Big(\sum_{i=1}^{2N}\frac{1}{p_i}(\sum_{j=1}^{C}\sum_{k=1}^{D}m_i\nabla_{\mathbf{Z}_{j,i}}^{\updownarrow}{}^{\top}\mathbf{X}_{i,k}^{\updownarrow})\Big)\\
&= \sum_{i=1}^{2N}\frac{p_i(1-p_i)}{p_i^2}Var\Big((\sum_{j=1}^{C}\sum_{k=1}^{D}\nabla_{\mathbf{Z}_{j,i}}^{\updownarrow}{}^{\top}\mathbf{X}_{i,k}^{\updownarrow})\Big)\\
&= \sum_{i=1}^{2N}\frac{1-p_i}{p_i}(\sum_{j=1}^{C}\sum_{k=1}^{D}\nabla_{\mathbf{Z}_{j,i}}^{\updownarrow}{}^{\top}{}^{2}\mathbf{X}_{i,k}^{\updownarrow}{}^{2}).
\end{aligned}
$$

$\square$

So that

$$Var(\nabla_{\mathbf{W}}) = \sum_{i=1}^{2N}(\frac{1}{p_i}-1)(\sum_{j=1}^{C}\nabla_{\mathbf{Z}_{j,i}}^{\updownarrow}{}^{\top}{}^{2})(\sum_{k=1}^{D}\mathbf{X}_{i,k}^{\updownarrow}{}^{2}) \tag{9}$$

$$= \sum_{i=1}^{2N}(\frac{1}{p_i}-1)\|\nabla_{\mathbf{Z}_{:,i}}^{\updownarrow}{}^{\top}\|^2\|\mathbf{X}_{i,:}^{\updownarrow}\|^2, \tag{10}$$

which proves.

### B.2  Proof of Activation Leverage Score in Sec. 4.2

we divide the matrix multiplication into the sum of $2N$ smaller multiplications:

$$\hat{\mathbf{I}}^{\updownarrow}\nabla_{\mathbf{Y}}^{\updownarrow}=\sum_{i=1}^{2N}\hat{\mathbf{I}}_{:,i}^{\updownarrow}\nabla_{\mathbf{Y}i}^{\updownarrow}=\sum_{i=1}^{2N}\hat{\nabla}_{\mathbf{Y}_i}, \tag{11}$$

where we define $\hat{\nabla}_{\mathbf{Y}_i}=\hat{\mathbf{I}}_{:,i}^{\updownarrow}\nabla_{\mathbf{Y}i}^{\updownarrow}$.

We assigns each $\nabla_{\mathbf{Y}_i}$ a probability $p_i \in [0, 1], i = 1, \cdots, 2N$, that satisfies $\sum_{i=1}^{2N} p_i = N$. We define random masks $m_i \sim \text{Bern}(p_i)$, and define $\tilde{\mathbf{M}} = \text{diag}\left(\frac{m_1}{p_1}, \ldots, \frac{m_{2N}}{p_{2N}}\right)$, and make an unbiased estimation:

$$\hat{\mathbf{I}}^{\updownarrow}\nabla_{\mathbf{Y}}^{\updownarrow} \approx \hat{\mathbf{I}}^{\updownarrow}\tilde{\mathbf{M}}\nabla_{\mathbf{Y}}^{\updownarrow} = \sum_{i=1}^{2N} \frac{m_i}{p_i}\nabla_{\mathbf{Y}i}^{\updownarrow}.$$

Define $\mathbf{M}^{\uparrow}$ to be the top-left $N \times N$ submatrix of $\mathbf{M}$ and $\mathbf{M}^{\downarrow}$ to be the bottom-right one, we have

$$\hat{\mathbf{I}}^{\updownarrow}\tilde{\mathbf{M}}\nabla_{\mathbf{Y}}^{\updownarrow} = s_{\uparrow}\mathbf{I}\tilde{\mathbf{M}}^{\uparrow}\nabla_{\mathbf{Y}}^{\uparrow} + s_{\downarrow}\mathbf{I}\tilde{\mathbf{M}}^{\downarrow}\nabla_{\mathbf{Y}}^{\downarrow},$$

In this case, $\mathbf{I}\tilde{\mathbf{M}}^{\uparrow}\nabla_{\mathbf{Y}}^{\uparrow}$ and $\mathbf{I}\tilde{\mathbf{M}}^{\downarrow}\nabla_{\mathbf{Y}}^{\downarrow}$ both only have parts of its rows being non zero, and the rest rows are zeros since they are discarded. Then, when we multiply it by $\hat{\mathbf{W}}$ , there are half of rows being zeros in $\mathbf{I}\tilde{\mathbf{M}}^{\uparrow}\nabla_{\mathbf{Y}}^{\uparrow}\hat{\mathbf{W}}$ and $\mathbf{I}\tilde{\mathbf{M}}^{\downarrow}\nabla_{\mathbf{Y}}^{\downarrow}\hat{\mathbf{W}}$. So there's no need to calculate them, and we successfully cut off half of the computation in this case.

Now focus on the variance that

**Proposition B.2.** *(LSS variance for activation gradient)*

$$\text{Var}\left[\sum_{i=1}^{2N} \hat{\mathbf{I}}_{:,i}^{\updownarrow}\nabla_{\mathbf{Y}i}^{\updownarrow}\right] = \sum_{i=1}^{2N} \frac{1-p_i}{p_i}\|\nabla_{\mathbf{Y}i}^{\updownarrow}\|^2.$$

*Proof.*

$$\begin{aligned}
Var(\nabla_{\mathbf{X}}) &= Var\left(\sum_{i=1}^{2N} \frac{1}{p_i}(m_i\hat{\mathbf{I}}_{:,i}^{\updownarrow}\mathbf{X}_i^{\updownarrow})\right) \\
&= Var\left(\sum_{i=1}^{2N} \frac{1}{p_i}(\sum_{j=1}^{C}\sum_{k=1}^{D} m_i\hat{\mathbf{I}}_{j,i}^{\updownarrow}\nabla_{\mathbf{Y}i,k}^{\updownarrow})\right) \\
&= \sum_{i=1}^{2N} \frac{p_i(1-p_i)}{p_i^2}Var\left((\sum_{j=1}^{C}\sum_{k=1}^{D}\hat{\mathbf{I}}_{j,i}^{\updownarrow}\nabla_{\mathbf{Y}i,k}^{\updownarrow})\right) \\
&= \sum_{i=1}^{2N} \frac{1-p_i}{p_i}(\sum_{j=1}^{C}\sum_{k=1}^{D}(\hat{\mathbf{I}}_{j,i}^{\updownarrow})^2(\nabla_{\mathbf{Y}i,k}^{\updownarrow})^2) \\
&= \sum_{i=1}^{2N}(\frac{1}{p_i}-1)(\sum_{j=1}^{C}(\hat{\mathbf{I}}_{j,i}^{\updownarrow})^2)(\sum_{k=1}^{D}(\nabla_{\mathbf{Y}i,k}^{\updownarrow})^2) \\
&= \sum_{i=1}^{2N}(\frac{1}{p_i}-1)\|\hat{\mathbf{I}}_{:,i}^{\updownarrow}\|^2\|\nabla_{\mathbf{Y}i}^{\updownarrow}\|^2 \\
&= \sum_{i=1}^{2N}(\frac{1}{p_i}-1)\|\nabla_{\mathbf{Y}i}^{\updownarrow}\|^2.
\end{aligned}$$

$\square$

In this way, the coefficient $c_i := \|\nabla_{\mathbf{Y}i}^{\updownarrow}\|$ is the *leverage score*.

## C  Experiments.

In this section, we present more details for experiments in Sec. 5.

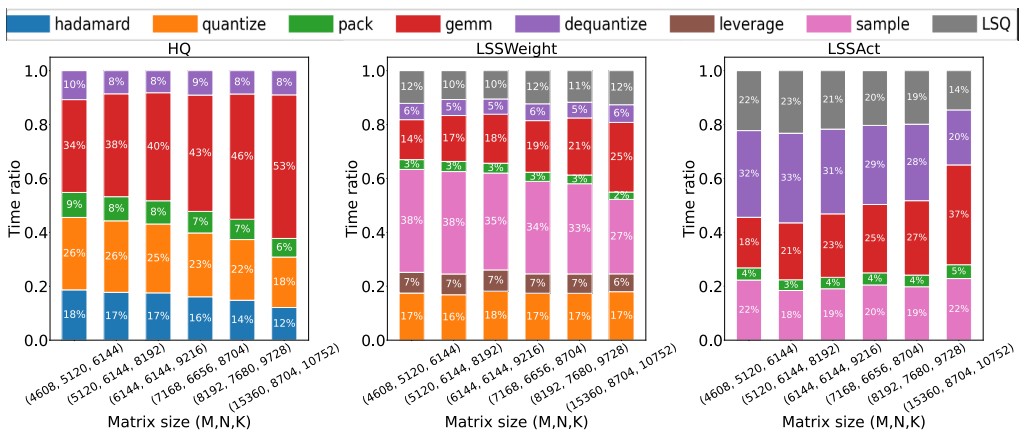

Figure 6: Time proportion for each part in `HQ-MM` and `LSS-MM` operator.

## C.1 Experiments setup

For the GLUE, QA, SWAG, and CONLL tasks, we implement our algorithm based on `https://github.com/huggingface/transformers`. For the machine translation task, we implement our algorithm based on `https://github.com/facebookresearch/fairseq`. For the ViT fine-tuning task, we implement our algorithm based on `https://github.com/jeonsworld/ViT-pytorch`. For the deit pretraining task, we implement our algorithm based on `https://github.com/facebookresearch/deit`.

We employed NVIDIA GeForce RTX 3090 for running most of the experiments, while the NVIDIA A40 was utilized to evaluate the performance of BERT-Large and ViT-L. Furthermore, we conducted runtime measurements using the NVIDIA T4, 3090, and A100 GPUs.

## C.2 GLUE results

In this section, we present the detailed result of fine-tuning the GLUE dataset on BERT-base-uncased and BERT-large-uncased.

On BERT-base, on STSB, SST2, QNLI, and QQP, HQ+LSS only has $< 0.5\%$ accuracy degradation. On the most challenging tasks CoLA and RTE, our accuracy degradation is much smaller compared to LSQ+LUQ. On QQP and MNLI, our method achieves $< 1.3\%$ degradation, while LSQ + LUQ has $\geq 1.8\%$ degradation. The trend is that the more difficult the task is, the more significant our advantage over LSQ+LUQ.

On BERT-large, the improvement is significant. On CoLA, QNLI, and MNLI, the accuracy improvement compared with LSQ+LUQ $> 30\%$. On other datasets like SST2 and QQP, the accuracy improvement is $> 10\%$. On RTE the accuracy improvement is $> 5\%$, and on STSB and MRPC the improvement is $> 3\%$.

We suspect that for those challenging tasks, there is more information stored in the outliers, which results in a larger gap between our method and LSQ+LUQ.

## C.3 More Granular Quantization Methods

In this section, in Table 5, we show that the more granular quantization methods, such as per-token quantization and per-channel quantization, or smoothing techniques, such as SmoothQuant, do not work under the 4-bit FQT setting. Meanwhile, combining these methods with HQ will not bring significant improvement.

We find that LSQ is beneficial for all of these more granular quantization methods under low-bit settings, which highlights the importance of LSQ. Meanwhile, we also notice that the smoothquant will even harm the result of LSQ when the bit-width is low. Our explanation is that the motivation of LSQ is to learn a trade-off between outliers and inliers, while smoothquant aims to sacrifice the

Table 3: GLUE results on BERT-base-uncased and BERT-large uncased. FP refers to full precision training, INT8 refers to INT8 training, LSQ + LUQ refers to learned step size quantization for forward and logarithmic unbiased quantization for backward, and HQ + LSS refers to Hadamard quantization for forward and leverage score sampling for backward.

| | | QUANTIZATION METHODS | | | |
|---|---|---|---|---|---|
| MODEL | DATASET | FP | INT8 | LSQ+LUQ | HQ+LSS |
| BERT-BASE | CoLA | $56.89_{0.64}$ | $56.15_{0.94}$ | $18.76_{3.58}$ | $\mathbf{52.46_{1.46}}$ |
| | STSB | $88.14_{0.73}$ | $87.05_{0.38}$ | $84.31_{0.29}$ | $\mathbf{87.77_{0.30}}$ |
| | RTE | $64.80_{1.26}$ | $62.27_{1.26}$ | $56.80_{0.92}$ | $\mathbf{62.45_{1.08}}$ |
| | MRPC | $88.61_{0.66}$ | $86.85_{0.76}$ | $86.23_{0.67}$ | $\mathbf{86.54_{0.83}}$ |
| | SST2 | $92.72_{0.06}$ | $92.37_{0.17}$ | $90.37_{0.46}$ | $\mathbf{92.49_{0.29}}$ |
| | QNLI | $91.52_{0.22}$ | $90.92_{0.24}$ | $87.33_{0.48}$ | $\mathbf{90.53_{0.23}}$ |
| | QQP | $91.09_{0.11}$ | $90.57_{0.05}$ | $89.26_{0.03}$ | $\mathbf{89.80_{0.05}}$ |
| | MNLI | $84.52_{0.22}$ | $84.10_{0.08}$ | $81.79_{0.18}$ | $\mathbf{83.59_{0.12}}$ |
| | MNLI-MM | $84.68_{0.20}$ | $84.49_{0.31}$ | $82.22_{0.33}$ | $\mathbf{83.75_{0.28}}$ |
| BERT-LARGE | CoLA | $60.33_{0.49}$ | $58.80_{1.52}$ | $0.00_{0.00}$ | $\mathbf{53.46_{1.17}}$ |
| | STSB | $87.59_{2.39}$ | $86.53_{0.20}$ | $83.08_{0.41}$ | $\mathbf{87.57_{0.78}}$ |
| | RTE | $71.12_{1.80}$ | $63.71_{1.26}$ | $53.06_{0.72}$ | $\mathbf{64.62_{0.78}}$ |
| | MRPC | $91.06_{0.28}$ | $87.57_{1.47}$ | $82.56_{0.59}$ | $\mathbf{87.62_{0.51}}$ |
| | SST2 | $93.98_{0.17}$ | $93.75_{0.63}$ | $83.94_{0.69}$ | $\mathbf{93.52_{0.40}}$ |
| | QNLI | $92.26_{0.05}$ | $92.29_{0.29}$ | $63.18_{13.10}$ | $\mathbf{91.53_{0.38}}$ |
| | QQP | $91.04_{0.63}$ | $90.71_{0.00}$ | $75.62_{12.44}$ | $\mathbf{90.77_{0.02}}$ |
| | MNLI | $86.71_{0.19}$ | $85.82_{0.08}$ | $33.42_{1.38}$ | $\mathbf{85.86_{0.10}}$ |
| | MNLI-MM | $86.41_{0.35}$ | $85.87_{0.14}$ | $33.54_{1.55}$ | $\mathbf{85.82_{0.07}}$ |

Table 4: Experiments on GPT2-base and Bert-large. Total time spent for epoch 1-5 are reported.

| | | TRAINING METHODS | | |
|---|---|---|---|---|
| MODEL | (HIDDEN_SIZE, INTERMIDIATE_SIZE, BATCH_SIZE) | FP16 | HQ+LSS | SPEEDUP |
| BERT-LARGE | (2560, 10240, 2048) | 15.094s | 18.949s | $\mathbf{-25.5\%}$ |
| | (4096, 16384, 1280) | 32.016s | 30.594s | $\mathbf{4.4\%}$ |
| | (5120, 20480, 960) | 47.418s | 39.482s | $\mathbf{16.7\%}$ |
| | (7680, 30720, 600) | 95.832s | 67.253s | $\mathbf{29.8\%}$ |
| | (8960, 35840, 480) | 128.441s | 83.388s | $\mathbf{35.1\%}$ |
| | (9600, 38400, 160) | 161.114s | 114.325s | $\mathbf{29.0\%}$ |
| | (12800, 51200, 100) | 326.265s | 255.966s | $\mathbf{21.5\%}$ |
| | (14400, 57600, 96) | 409.291s | 346.354s | $\mathbf{15.3\%}$ |
| GPT2-BASE | (2560, 10240, 1536) | 17.253s | 22.037s | $\mathbf{-27.7\%}$ |
| | (4096, 16384, 960) | 35.937s | 35.694s | ~ |
| | (5120, 20480, 768) | 52.723s | 46.548s | $\mathbf{11.7\%}$ |
| | (7680, 30720, 260) | 113.855s | 92.548s | $\mathbf{18.7\%}$ |
| | (8960, 35840, 200) | 150.680s | 114.881s | $\mathbf{23.8\%}$ |
| | (9600, 38400, 180) | 172.182s | 126.540s | $\mathbf{26.5\%}$ |
| | (12800, 51200, 112) | 320.757s | 236.433s | $\mathbf{26.3\%}$ |

precision of inliers in order to exactly maintain the information of outliers. When the bitwidth is high, this is not a problem, since there are still enough bits to quantize the inliers. But when the bitwidth is low, such sacrifice will cause severe problems since the inlier information is discarded.

## C.4   Large Language Model Operator Speed

In this section, we show that our hardware-friendly INT4 training method can really accelerate the training process on Large Language Models. We run distributed training on a system of 8 A100 cards

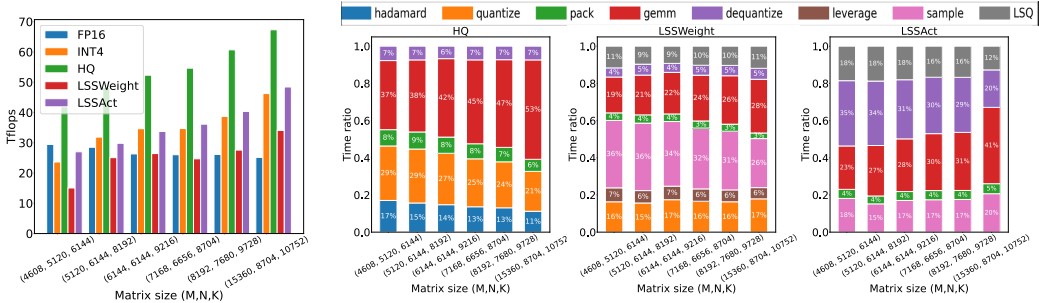

Figure 7: Real quantization performance on Nvidia T4.

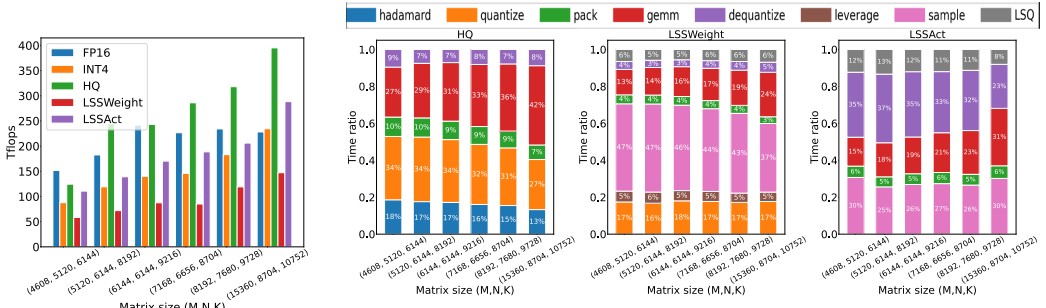

Figure 8: Real quantization performance on Nvidia A100.

Table 5: Comparison of different quantization methods, quantize the activation and weight into the same bit-width from 2 to 8. Per-token refers to quantize activation per-token, while Per-channel refers to quantize weight per-channel.

| quantization methods | Quantize Bits | | | | | | |
|---|---|---|---|---|---|---|---|
| | 2 | 3 | 4 | 5 | 6 | 7 | 8 |
| Per-tensor | 0 | 0 | 0 | 0 | 0 | 50.2 | 54.6 |
| Per-token | 0 | 0 | 0 | 0 | 31.4 | 52.8 | 56 |
| Per-channel | 0 | 0 | 0 | 0 | 0 | 51.9 | 56.7 |
| smoothquant | 0 | 0 | 0 | 0 | 0 | 49.4 | 57.7 |
| Per-token + Per-channel + smoothquant | 0 | 0 | 0 | 0 | 40.7 | 55.7 | 56.7 |
| LSQ | 0 | 9.16 | 24.2 | 37.3 | 39.6 | 45.3 | 51.4 |
| Per-token + LSQ | 0 | 15.3 | 27.8 | 31.6 | 42.9 | 46 | 54.4 |
| Per-channel + LSQ | 0 | 8 | 23.9 | 29.3 | 40 | 45.5 | 50.7 |
| smoothquant + LSQ | 0 | 0 | 0 | 0 | 49.6 | 54.9 | 57 |
| Per-token + Per-channel + smoothquant + LSQ | 0 | 0 | 0 | 0 | 28.8 | 52.4 | 55.2 |
| HQ | 0 | 45.2 | 54.6 | 54.2 | 56.5 | 57.4 | 58.4 |
| HQ + Per-token + Per-channel | 0 | 48.4 | 54.1 | 54.9 | 55 | 56 | 56 |
| HQ + Per-token + Per-channel + smoothquant | 0 | 0 | 46.6 | 54.9 | 55.9 | 55.8 | 56.5 |

and our implementation uses distributed data parallel training with zero-3, gradient checkpointing, and optimizer offloading.

We experimented with two architectures: BERT-Large and GPT2-base. We vary the network width and batch size to make full utilization of the GPU memory and show the end-to-end performance for fine-tuning these models on the SuperGLUE RTE dataset in Table 4.

## C.5 More experiments on Operator Speed

**Time proportion** We examine the proportion of time for each part of computation in `HQ-MM` and `LSS-MM` operator in Fig. 6 when the shapes of input matrices vary. In HQ, hadamard means multiplying the input matrix with the Hadamard matrix, pack means packing input data into INT4 data, gemm means the matrix multiplication of two INT4 matrices. In LSSWeight, quantize corresponds to the quantization of higher and lower 4-bit, leverage means computing leverage score, sample means sample out rows/columns given the leverage score, dequantize is the process of dequantizing INT data back into FP16 data, and LSQ is the backpropagation process of LSQ method. In LSSAct, we ignore quantize and leverage process, using the same value as LSSWeight for saving time, other processes share the same meaning with LSSWeight. Note that our implementation is not fully optimized, and optimizations like operator fusion can further improve the performance.

**Operator Speed on more GPUs** On an Nvidia RTX 3090 GPU with a Cuda capability of sm_86., we show the comparison of FP16 MM, HQ, and LSS operators in Section 5.3 as well as time proportion in each operator in Figure. 6. We also adjust our hardware implementation and test its performance on Nvidia T4 GPU and Nvidia A100 GPU, which have Cuda capability of sm_75 and sm_80 , respectively. The result is shown in Fig. 7 and Fig. 8.

