# OpenReview forum: "Training Transformers with 4-bit Integers"
_NeurIPS.cc/2023/Conference — NeurIPS 2023 poster_

### Official Review · Reviewer_XvWB · 2023-07-06

**Soundness:** 3 good
**Presentation:** 3 good
**Contribution:** 2 fair
**Rating:** 4
**Confidence:** 4

**Summary:**

This paper presents a 4-bit training method mainly to speed up the training process. In the forward propagation, Hadamard quantizer is proposed to suppress the outliers whereas structure sparsity and bit splitting are used for quantization of gradients. It has been shown that the proposed training method can successfully train transformer models across different datasets with some performance degradation w.r.t. the baseline.

**Strengths:**

-- The main strength of this paper is theoretical analysis of their proposed quantization method for both forward and backward steps.
-- Use of different models and tasks for evaluation purposes is another strength of this paper.

**Weaknesses:**

-- The main concern of mine is the limited speedup of this paper. The main goal of quantized training in this paper is to speed up the computations and not to reduce the memory. However, the authors have only shown up to 35% speedup w.r.t. the baseline (FP16). First, there is no comparison with Int8 training speed. It's not clear why I should choose Int4 over Int8 while I know the accuracy of Int8 is better than Int4. Second, the speedup amounts are only measured across specific configurations which are not the ones used for evaluation purposes in Table 1. Measuring the training time for the tasks listed in Table 1 can make the contribution of this paper clear.
-- Fig. 4 and Fig. 5 are hard to read. Again, why not simply reporting the speedup for each task in Table 1.
-- I have a hard time understanding Table 3. What does "epoch 1-5" mean? Which dataset was used for the measurement? What is the accuracy performance?

**Questions:**

In summary, the idea of Int4 training to accelerate the process is interesting. The authors have shown that Int4 training can obtain a comparable accuracy w.r.t. the baseline and Int8 with around ~1% to ~2% degradation, which is fine. However, the speedup amount is not provided for the benchmarks used for accuracy evaluation in Table 1. Given the above explanations, I have the following questions:
1) What is the speedup for tasks listed in Table 1?
2) How do you compare your method w.r.t. Int8 in terms of training speed? In other words, why should I select Int4 over Int8?
3) How was the accuracy performance measured? Did you report the accuracy in Table 1 using quantized inference (Int4) or full-precision?
4) Please provide more details for Table 3.

---

> ### Author Rebuttal · Authors · 2023-08-09
>
> We thank you for your comments and feedback. Below we provide a point-to-point response to all comments.
>
> **W1**: limited speedup of this paper
>
> **A1**: We would like to emphasize that our main contribution is the first INT4 training **algorithm**. The implementation is only a small fraction of our contribution, and our paper should not be evaluated mainly based on the implementation.
> As we stated in the paper, our implementation is only a prototype, **it does not fully reflect the potential of our INT4 training algorithm**. Our implementation has much room to improve, such as operator fusion. However, the engineering efforts for a heavily optimized INT4 training system exceed the scope of a research paper on the first INT4 training algorithm.
>
> INT4 training is an extremely difficult task, with challenges ranging from numerical format, optimization, model architecture, and software and hardware implementation.
> It is impractical to solve all the challenges with a single research paper. For example, there already exists INT8[10]/FP8[11] training algorithms in 2018, but the accleration potential of FP8 for training is realized 4 years later in Nvidia's H100 GPUs with Transformer Engine in 2022, and how to use INT8 to accelerate training is still an active area of research.
>
> Among the challenges of INT4 training, the first question to answer is, does there exists an INT4 training **algorithm**, which can achieve good accuracy by utilizing mainly low-precision arithmetic? Our work gives an initial affirmative answer to this question. This is the main contribution of our research. Only when we have an algorithm, people can then conduct research to optimize the actual software and hardware implementation and optimize the algorithm to be more accurate and software/hardware friendly. Actually, most research papers on low-precision training (e.g., [6,7,10,11]) only do simulated quantization, without having an actual implementation. While we do provide an implementation, it is only a prototypical one without heavy optimizations, as we stated in the paper. The implementation optimization itself can be a worthy research, and we leave it as a future work.
>
> **W2**: No comparison with INT8 training speed
>
> **A2**: To the best of our knowledge, there are currently no open-source implementations of INT8 training for transformers in the existing literature.
> Most low-precision training works simply do simulated quantization, without a real implementation to actually realize the speedup.
> There exists some INT8 training work [1-5] that reports speedup, but none of them are open-source. Particularly, [1-4] focus on INT8 training of CNN-based models, but none of them are open-source, [5] has open-sourced their INT8 training Python code without the actual hardware implementation.
>
> **Q3**: What is the speedup for tasks listed in Table 1?
>
> **A3**: Our implementation cannot yet achieve speedup for tasks listed in Table 1, due to their small model size. However, as argued before, our implementation still has much room for optimization, and implementation is not the main focus of this paper.
>
> **Q4**: How do you compare your method w.r.t. INT8 in terms of training speed? In other words, why should I select Int4 over INT8?
>
> **A4**: Our INT4 training algorithm is superior than an INT8 one regarding to time complexity.
> Consider multiplying a matrix of $N\times C$ with another matrix of $C\times D$. Our INT4 MM takes $2NCD$ int4 multiplication-and-additions (MACs), while an INT8 MM takes $2NCD$ int8 MACs. The amount of FP16 operations required by our INT4 MM is $O(2^k N (D+C))$, where a typical value of $k=5$ is the group size. As matrix sizes grow large, the proportion of overhead diminishes, and the speedup of INT4 over INT8 should approach 2x. So we believe that INT4 is ultimately more promising than INT8. The speedup potential might not be realized in this single paper, as discussed earlier.
>
> Another reason for favoring INT4 over INT8 lies in the model's inference speed. The model trained by our algorithm can perform INT4 inference, while the model trained by an INT8 algorithm can only use INT8 for inference. At inference time, only the fast HQ-MM operator is used. As reported in Table 4, this leads to a 2.44-3.50 speedup compared to FP16, which should also be faster than INT8. For the inference phase, we can further optimize the implementation by precomputing the process of Hadamard transformation and quantization of the weight and achieving about another 20\% speed acceleration.
>
> **Q5**: How was the accuracy performance measured? Did you report the accuracy in Table 1 using quantized inference (Int4) or full precision?
>
> **A5**: In Table 1, the "HQ+LSS" accuracy is measured using INT4 quantized inference by applying the hadamard quantizer. In the training phase, the weight is quantized to INT4 using our Hadamard Quantizer, so it is suitable for INT4 inference. And in a real application, INT4 inference will bring acceleration. So we report the accuracy using INT4 quantized inference in Table 1.
>
> **Q6**: Please provide more details for Table 3.
>
> **A6**: In Table3, we run distributed training on a system of 8 A100 cards. Our implementation uses distributed data-parallel training with zero-3, gradient checkpointing, and optimizer offloading.
>
> We experimented with two architectures: BERT-Large and GPT-2. We vary the network width and batch size to make full utilization of the GPU memory.
>
> By comparing our implementation with the baseline fp16 model, we can see that the end-to-end performance for fine-tuning these models is more efficient in our case, with an up to 35\% speedup when the parameters are appropriate.
>
> We are grateful for the time and effort you have invested in evaluating our submission. We are more than willing to address any further questions or concerns you may have.
>
> The references are listed in the global rebuttal section due to the rebuttal length restriction.

---

> > ### Comment · Reviewer_XvWB · 2023-08-18
> > **Rebuttal Comments**
> >
> > Thank you for the clarification and responses to my comments. My main concern here is the speedup gain from the proposed algorithm. I can understand that INT4 training is a hard task but the main goal of such a method (that results in an accuracy degradation without any memory saving during the training process) is to speed up the training process. Therefore, it's expected to show a superior speed compared to existing INT8 training methods. If there is no open-source INT8 training method, then you can at least compare with INT8 inference methods such as I-BERT which is available on HuggingFace.

---

> > > ### Author Response · Authors · 2023-08-21
> > >
> > > Thanks for the suggestion.
> > > Unfortunately, the open-sourced versions of I-BERT (including the official codebase and HuggingFace) are based on fake quantization, and they cannot provide any speedup. Their TensorRT implementation is not open-sourced (see [1, 2] for reference).
> > >
> > > To address reviewer's concerns, we compare the inference speed of our algorithm with I-BERT by comparing the speedup numbers reported in its original paper [3].
> > > Following the I-BERT paper, we compare the speedup of integer-based inference algorithms relative to a FP32 baseline on an Nvidia T4 GPU on the bert-base and bert-large models, and test with sequence lengths of 128 and 256. While I-BERT only reported speedup numbers for batch sizes \{1, 2, 4, 8\}, we test the speedup for batch sizes ranging from 1 to 1024 to better reflect the performance of throughput-oriented scenarios (such as a cloud language model service provider).
> > >
> > > Table 1: Speedup of integer-based inference algorithms compared to a FP32 baseline on an Nvidia T4 GPU. Sequence Length=128
> > >
> > > | batch size                  | 1    | 2    | 4    | 8    | 16   | 32   | 64   | 128  | 256  | 512  | 1024 |
> > > | --------------------------- | ---- | ---- | ---- | ---- | ---- | ---- | ---- | ---- | ---- | ---- | ---- |
> > > | bert-base speedup (I-BERT)  | 2.42 | 3.36 | 3.39 | 3.31 | -    | -    | -    | -    | -    | -    | -    |
> > > | bert-base speedup (ours)    | 0.17 | 0.25 | 0.46 | 0.87 | 1.22 | 2.07 | **3.61** | **3.57** | **4.50** | **4.54** | **4.92** |
> > > | bert-large speedup (I-BERT) | 3.2  | 4    | 3.98 | 3.81 | -    | -    | -    | -    | -    | -    | -    |
> > > | bert-large speedup (ours)   | 0.22 | 0.43 | 0.81 | 1.46 | 2.13 | 3.34 | **4.28** | **4.81** | **5.40** | **6.08** | **6.48** |
> > >
> > > Table 2: Speedup of integer-based inference algorithms compared to a FP32 baseline on an Nvidia T4 GPU. Sequence Length=256
> > >
> > > | batch size                  | 1    | 2    | 4    | 8    | 16   | 32   | 64   | 128  | 256  | 512  | 1024 |
> > > | --------------------------- | ---- | ---- | ---- | ---- | ---- | ---- | ---- | ---- | ---- | ---- | ---- |
> > > | bert-base speedup (I-BERT)  | 3.11 | 2.96 | 2.94 | 3.15 | -    | -    | -    | -    | -    | -    | -    |
> > > | bert-base speedup (ours)    | 0.25 | 0.54 | 0.95 | 1.53 | 1.94 | **3.20** | **3.76** | **4.00** | **4.15** | **4.15** | **4.14** |
> > > | bert-large speedup (I-BERT) | 3.19 | 3.51 | 3.37 | 3.4  | -    | -    | -    | -    | -    | -    | -    |
> > > | bert-large speedup (ours)   | 0.45 | 0.86 | 1.47 | 2.44 | 2.67 | **3.99** | **4.87** | **5.24** | **5.11** | **5.41** | OOM  |
> > >
> > >
> > > While I-BERT's speedup numbers seems to be insensitive to the batch size and sequence length, our speedup increases with the batch size. I-BERT shows up to 3.98x speedup for smaller batch size, while our algorithm can achieve higher speedup for batch sizes higher than 64, and eventually gives a speedup of 6.49x for a sequence length of 256 and a batch size of 1024.
> > >
> > > - For bert-base, our method shows an inference speed improvement of 3.57-4.92 times compared to FP32 for batch sizes larger than 64. In comparison, I-BERT achieved an inference speed improvement of 2.42-3.39 times compared to FP32 when the batch size is small.
> > > - For bert-large, our method shows an inference speed improvement of 4.81-6.48 times compared to FP32 for batch sizes larger than 64. In comparison, I-BERT achieved an inference speed improvement of 3.20-4.00 times compared to FP32 when the batch size is small.
> > >
> > > Therefore, our algorithm can potentially achieve higher throughput than I-BERT.
> > > Again, we would like to emphasize that our implementation does not fully reflect the potential of our INT4 training algorithm due to limited optimization. Additionally, I-BERT speedup numbers are achieved with the highly-optimized TensorRT, while our implementation is based on PyTorch.
> > >
> > > References:
> > >
> > > [1] https://github.com/kssteven418/I-BERT/issues/2
> > >
> > > [2] https://github.com/huggingface/transformers/issues/11312
> > >
> > > [3] Kim, Sehoon, et al. "I-bert: Integer-only bert quantization." International conference on machine learning. PMLR, 2021.

---

### Official Review · Reviewer_au4S · 2023-07-07

**Soundness:** 3 good
**Presentation:** 4 excellent
**Contribution:** 3 good
**Rating:** 6
**Confidence:** 3

**Summary:**

The authors propose 4-bit training methods for Transformers. It first proposes a Hadamard quantizer with activation outliers issue being solved by Hadamard Transformation. Optimization for backpropagation is proposed by leveraging gradient sparsity with bit splitting and score sampling. The proposed 4-bit training methods achieve competitive accuracy on NLP, Machine translation, and Image classification.

**Strengths:**

* Achieves competitive or superior accuracy compared with existing works on 4-bit training.
* Optimize backpropagation with hardware-efficient structural sparsity.
* Hardware-friendly INT4 training method is runnable on GPUs with up to 2.2 times faster than the FP16 MM.

**Weaknesses:**

* The activation outlier handling in Transformers is not new, as in SmoothQuant. And the comparison is lacking in the paper. Are those techniques not able to be used in 4-bit training? And what is the efficiency comparison between those methods when handling outlier activations?
* Could you elaborate more on why the proposed methods cannot be applied to Conv layers? What is the bottleneck?

**Questions:**

* It would be helpful if you could explain why the speed is not comparable with the baseline in Fig.5 when the hidden layer size is small.

**Limitations:**

* Conv cannot be supported
* INT4 training on extremely large models remains to be an open question.

---

> ### Author Rebuttal · Authors · 2023-08-09
>
> We thank you for your comments and feedback. Below we provide a point-to-point response to all comments.
>
> **Q1**: The comparison with SmoothQuant
>
> **A1**: There might be some misunderstandings.
> It's important to note that SmoothQuant is primarily designed for **INT8 inference** and not designed for **INT4 training**. INT4 is significantly more challenging than INT8. Furthermore, SmoothQuant is a post-training quantization method, and the weight can be quantized in a pre-processing stage. On the other hand, in our scenario, all the tensors must be quantized on-the-fly in each iteration. Furthermore, it is unclear whether post-training quantization methods can produce stable gradients for training. Consequently, a direct comparison between our approach and SmoothQuant might not be appropriate. Nonetheless, we have conducted a thorough comparative analysis with SmoothQuant, and the results are shown in Appendix C.3 as well as Table 4.
>
> In Appendix.C.3, we show that SmoothQuant always fails (0\% accuracy) in the INT4 training setting. We also show that even if we combined SmoothQuant with the more granular quantization methods, such as per-token quantization and per-channel quantization, it still does not work under the 4-bit FQT setting. Meanwhile, combining these methods with HQ will not bring significant improvement.
>
> Intuitively, our HQ is stronger than SmoothQuant.
> Our explanation is that the motivation of LSQ is to learn a trade-off between outliers and inliers, while smoothquant aims to sacrifice the precision of inliers in order to exactly maintain the information of outliers. When the bitwidth is high, this is not a problem, since there are still enough bits to quantize the inliers. But when the bitwidth is low, such sacrifice will cause severe problems since the inlier information is discarded.
>
> **Q2**: why the proposed methods cannot be applied to Conv layers
>
> **A2**: Thank you for bringing up this concern.
> INT4 training is an extremely challenging open problem, we leverage the specific structure of matrix multiplications to make this possible.
>
> Specifically, we leverage the sparsity property of transformer gradients. For transformers, the back propagation is multiplying a [batch size * sequence length, hidden size] gradient matrix with an activation / weight matrix. The gradient matrix has some almost-zero rows, so we can save computation by dropping these rows for matrix multiplication. However, Conv layers convolve the [batch size, height, width, channel] gradient map with another kernel. Even if the gradient map have spatial sparsity (almost zero for some locations), such sparsity cannot be easily leveraged to reduce the amount of computation.
>
> **Q3**: INT4 training on extremely large models remains to be an open question.
>
> **A3**: We appreciate the constructive criticism and insightful observations regarding the technical challenges and scalability of our method. We recognize that even INT8 training of large models like OPT-175B is an open problem (see [9] Appendix.C.3). So our work is actually a pioneer work on fully quantized training, since we focus on an even more aggressive INT4 setting.
>
> We are grateful for the time and effort you have invested in evaluating our submission. We are more than willing to address any further questions or concerns you may have.
>
> The references are listed in the global rebuttal section due to the rebuttal length restriction.

---

> > ### Author Response · Authors · 2023-08-21
> >
> > Q4: It would be helpful if you could explain why the speed is not comparable with the baseline in Fig.5 when the hidden layer size is small.
> >
> > A4: Thank you for bringing up this concern.
> > When the hidden layer size is small, the overhead of the element-wise quantization operator can surpass the reduction of the cost of matrix multiplication, so our INT4 training algorithm might be slower than the FP16 baseline.
> > As the hidden layer size increases, the time proportion of matrix multiplications (MMs) using INT4 precision becomes dominant, resulting in improved speed up relative to the baseline. Note that our INT4 MM is only a prototype implementation, and more optimization (such as operator fusion) could reduce the overhead and further improve the speed.

---

### Official Review · Reviewer_7XeV · 2023-07-07

**Soundness:** 3 good
**Presentation:** 3 good
**Contribution:** 2 fair
**Rating:** 5
**Confidence:** 4

**Summary:**

The research paper presents a framework for training transformer-based neural networks using 4-bit integers, offering enhanced computational and memory efficiency.
The proposed method includes a dedicated Hadamard quantizer for forward propagation, designed to suppress outliers which typically cause degradation of model accuracy. The Hadamard quantizer works by transforming the activation matrix, spreading outlier information across nearby entries in the matrix, and reducing the numerical range of outliers.
In backpropagation, the authors exploit the structural sparsity of activation gradients. They note that a few token gradients are extremely large, while most are much smaller, even below the quantization residuals of larger gradients. To make efficient use of computational resources, the authors propose a technique called bit splitting. This technique divides each token's gradient into two parts: the higher 4 bits and the lower 4 bits. They then identify the most informative gradients using a technique known as leverage score sampling, an importance sampling technique in the field of RandNLA.
The combination of these techniques results in an algorithm that employs 4-bit integer MMs for all linear operations in transformers. This algorithm has been evaluated and found to perform competitively across a wide range of tasks, including natural language understanding, question answering, machine translation, and image classification.

**Strengths:**

- Unlike many other low-precision training methods, this proposed method does not require custom numerical formats and is compatible with contemporary hardware, such as GPUs. This opens the potential for widespread adoption and usability.
- The proposed method demonstrated competitive or superior accuracy across a range of tasks, compared to existing 4-bit training methods. This indicates the practical effectiveness of the method in diverse applications.
- The authors have developed specialized quantization methods for both forward and backpropagation. These techniques manage outliers and leverage the structural sparsity of activation gradients, maintaining model accuracy despite the ultra-low numerical precision.

**Weaknesses:**

My main concern is that the paper chose to compare the performance and computation complexity of their approach with FP16 rather than examining its efficiency against INT8 or other INT4 alternatives. This could potentially limit the depth and breadth of the performance analysis.

The key contributions of this research are primarily focused on the implementation of a Hadamard quantizer and bit splitting techniques. Both techniques leverage INT4 matrix multiplication (MM). However, the HQ-MM has the 4 steps procedure to leverage the efficiency of INT4 computation and the bit-splitting technique deploys two INT4 MMs as a replacement for one INT8 MM. This can potentially induce more overhead than simply utilizing INT8, particularly considering that INT4 does not straightforwardly translate to half the computational, memory access, or energy costs associated with INT8 computations.

While the aim of utilizing INT4 computations is to increase efficiency, it's crucial to recognize that the benefits of moving from INT8 to INT4 are not always linear. According to a report by NVIDIA, INT4, in an optimal scenario involving Convolutional Neural Network (CNN) inference in image classification tasks, resulting in a throughput improvement of approximately 59% with a negligible accuracy loss (around 1%) on NVIDIA T4. Meanwhile, on TITAN RTX, the speedup was about 52%. These improvements, although significant, don't necessarily equate to halving the computational resources or doubling the performance, and this is also applied to the transformer-based models scenario of this paper.

Hence, a more thorough comparative study involving INT8 and other INT4 counterparts could potentially provide a more nuanced and comprehensive understanding of the computational advantages and trade-offs of the proposed techniques. Additionally, if there are any misinterpretations or inaccuracies in my understanding of the paper, I would appreciate it if you could provide corrections to ensure my perspective aligns accurately with the content and intentions of the research.


**Questions:**

The concept of using a Hadamard matrix to decompose matrix multiplication is a thoughtful and fitting approach in low-bit quantization scenarios, especially for extreme cases such as binary and ternary methods. The paper meticulously analyzes the time complexity of the Hadamard Quantizer Matrix Multiplication (HQ-MM) procedure in Section 3.3, and in Appendix A.5, it further elucidates on how to select an appropriate Hadamard matrix size. I am intrigued to learn more about the finer details of this procedure.

Given the significant shift in distribution during training, it is indeed crucial to reconfigure both the Hadamard matrix size and the Learned Step Size Quantization (LSQ) step size. This brings up a few questions: Is the re-initialization process executed only once, or is it a recurring process? Does the system stabilize after approximately two epochs? Furthermore, is X representative of the activation of a batch of input data? These details will enhance the understanding and application of the proposed methodology.



**Limitations:**

Implementing advanced quantization techniques such as the Hadamard quantizer and bit splitting may be technically challenging and potentially difficult to integrate into existing neural network training pipelines.
The method exhibits limitations in its scalability, as it is not equipped to support exceedingly large models like OPT-175B. This underscores a broader, unresolved issue in the realm of machine learning, where even the application of INT8 training to these substantial models remains an open-ended problem.

---

> ### Author Rebuttal · Authors · 2023-08-09
>
> We thank the reviewer for the acknowledgment of the potential and effectiveness of our work and the detailed constructive comments. Below we provide a point-to-point response to all comments.
>
> **W1**: compare the performance and computation complexity with FP16 rather than examining its efficiency against INT8 or other INT4 alternatives
>
> **A1**: There might be some misunderstandings. There are many INT8 and INT4 implementations for **inference**, but to the best of our knowledge, there are currently no open-source implementations of INT8 / INT4 **training** for transformers in the existing literature.
> Most low-precision training works simply do simulated quantization, without a real implementation to actually realize the speedup.
> There exists some INT8 training work [1-5] that reports speedup, but none of them are open-source.
>
> To the best of our knowledge, we are the first work that performs INT4 fully-quantized training. So there are no other INT4 alternatives.
>
> **W2**: speedup potential of INT4 comparing to INT8
>
> **A2**: We agree with the reviewer that the benefit of reducing numerical precision is not always linear, and there are still many implementation challenges to consider before INT4 training can be really used in commodity training systems.
>
> Regarding the overhead, indeed, INT4 MM can induce more overhead than simply utilizing INT8. However, as the additional operations of HQ-MM and LSS-MM have lower time complexity $O(2^k N (D+C))$ compared to $O(NCD)$ (where $k=5$ is the group size, $N$ is the batch size, $D, C$ are a number of neurons), the proportion of overhead decreases as the matrices grow large. So we believe that INT4 is ultimately more promising than INT8.
>
> We would like to emphasize that the implementation is only a small fraction of our contribution, and our paper should not be evaluated mainly based on the implementation. INT4 training is an extremely difficult task, with challenges ranging from numerical format, optimization, model architecture, and software and hardware implementation.
> It is not likely to solve all the challenges with a single research paper.
>
> Among the challenges, the first question to answer is, does there exists an INT4 training **algorithm**, which can achieve good accuracy by utilizing mainly low-precision arithmetic? Our work gives an initial affirmative answer to this question. This is the main contribution of our research.  Actually, most research papers on low-precision training (e.g., [6, 7], [10, 11]) only do simulated quantization, without having an actual implementation. While we do provide an implementation, it is only a prototypical one without heavy optimizations, as we stated in the paper. The implementation optimization itself can be a worthy research, and we leave it as a future work.
>
> **Q3**: Finer details of Hadamard Quantizer
>
> **A3**: We search the Hadamard group size among the exponential of 2, from 1 to 1024. Before the actual training, we pass a "probing batch" to the model and use it to search for the optimal Hadamard group size and LSQ step size. For each of the linear layers, we search the group size by quantizing the activation and weight. We select the group size with the lowest $MSE(\bar{X}, X) * MSE(\bar{W}, W)$, which is described in Appendix A.5.
>
> **Q4**: Details about the re-initialization process.
>
> **A4**: For the re-initialization process, we usually repeatedly do this at the end of each epoch for the first few epochs.
> Then, the Hadamard group size is frozen in the rest of the training, and the LSQ step size is learned jointly with model parameters.
> We observe that $\frac{2mean(x)}{Q_p}$ (which is how we calculate the initial value of LSQ step size) changes dramatically during the first few epochs, then it stabilizes.
>
> A rule of thumb is, the system definitely stabilizes when the accuracy is relatively high.
> For the WMT experiment, we stop to re-initialize at the second epoch, since the BLEU score is relatively high at the second epoch (about 21, while the final BLEU score after 30 epochs is 25.5). However, for the deit-small pretraining experiment, we re-initialize it at the end of the 10th epoch (we train it for 90 epochs in total), since at the end of the second epoch, the accuracy is not high, but at the end of the 10th epoch, the accuracy is relatively high.
> We will revise our final version to include these details.
>
> **Q5**: Is X representative of the activation of a batch of input data?
>
> **A5**: We randomly choose the batch of input data for initialization. But we do not observe too much difference when we change the data we used for initialization. For the hadamard group size, possibly it will change slightly when we change the data used for initialization (from 64 to 32, for example), but this will not bring too much difference since it can still reduce the outlier magnitude. And for the LSQ step size, since it continually changes during the training, as long as the initial value is not very far away from the optimal step size, which data used for initialization will not affect the final result. We conducted each experiment with three different seeds, and their results are similar, further validating our viewpoint.
>
> **Q6**: Limitations in its scalability, as it is not equipped to support exceedingly large models like OPT-175B.
>
> **A6**: We appreciate the constructive criticism and insightful observations regarding the technical challenges and scalability of our method. We recognize that even INT8 training of large models like OPT-175B is an open problem (see [9] Appendix.C.3). So our work is actually a pioneer work on fully quantized training, since we focus on an even more aggressive INT4 setting.
>
> We are grateful for the time and effort you have invested in evaluating our submission. We are more than willing to address any further questions or concerns you may have.
>
> The references are listed in the global rebuttal section due to the rebuttal length restriction.

---

> > ### Comment · Area_Chair_PVy3 · 2023-08-18
> >
> > Dear reviewer,
> >
> > The discussion with authors is closing soon, please review the rebuttal to see if the authors have addressed your concerns, and acknowledge to the authors that you have read their response.
> >
> > Thanks

---

> > ### Comment · Reviewer_7XeV · 2023-08-22
> >
> > Dear Authors, thank you for your detailed response. Most of my concerns have been addressed. I appreciate your efforts in clarifying the points raised.

---

### Official Review · Reviewer_jhrg · 2023-07-07

**Soundness:** 3 good
**Presentation:** 3 good
**Contribution:** 3 good
**Rating:** 6
**Confidence:** 5

**Summary:**

This paper deals with 4-bit training (fine-tuning) of transformers. For forward propagation, a Hadamard quantizer is proposed to solve the problem of outliers. For backpropagation, the authors leverage the structural sparsity of gradients by proposing bit splitting and leverage score sampling
techniques to quantize gradients accurately. Experiments show up to 35.1% speedup for 4-bit training with

**Strengths:**

1. The proposed Hadamard quantizer is simple while useful for transformer quantization with outliers.
2. The authors illustrate the structural sparsity of gradients and propose bit splitting (BS), which splits a full-precision matrix as higher and lower 4 bits. The similar idea is used in quantization for inference. This is the first to use it for training.
3. The experiments are extensive. The training speed on GPUs are reported.

**Weaknesses:**

1. The proposed method introduce extra FP16 computations. The BS with selection needs data rearrangement, which could slow down the speed.
2. The title is somewhat over-claim. The experiments are mainly about fine-tuning (FT). It seems that the proposed method works for fine-tuning (FT) of a converged model. For pre-training (PT), there is large accuracy loss.
3. The speed-up of up to 35.1% is misleading. It is better to use the average speed-up in the abstract.

**Questions:**

Please see the weaknesses.

---

> ### Author Rebuttal · Authors · 2023-08-09
>
> We thank the reviewer for the acknowledgment of our contributions and the careful constructive comments.  Below we provide a point-to-point response to the reviewer's questions.
>
> **Q1**: The proposed method introduces extra FP16 computations. The BS with selection needs data rearrangement, which could slow down the speed
>
> **A1**: Thanks for the insightful comment. While it is true that the proposed method involves some additional FP16 computations to calculate the leverage score, it is important to note that the time complexity of additional FP16 computations is lower than that of the original matrix multiplication.
> For the FP16 operation used in the Hadamard Quantizer, as discussed in Section 3.3, we show that the amount of FP16 computation is $O(2^kN(D + C))$, which is much cheaper than the original $O(NCD)$ matrix multiplication(where a typical value of $k=5$ is the group size, $N$ is the batch size, $D, C$ are a number of neurons). Therefore, the overhead is negligible as matrices grow large. In practice, the FP16 computation only takes 12% $\sim$ 18% time in our HQ-MM operator, as shown in Fig. 6 in the appendix.
>
> It is true that the data rearrangement needed for the BS with selection could slow down the speed. Again, the time complexity of the data rearrangement step is $O(NC + ND)$, while the time complexity of the matrix multiplication is $O(NCD)$. So our method can still be attractive when the matrices are large. As shown in Fig. 4, our INT4 operators can be up to 2.2x faster than FP16.
>
>
> **Q2**: The title is somewhat over-claim. The experiments are mainly about fine-tuning (FT). It seems that the proposed method works for fine-tuning (FT) of a converged model. For pre-training (PT), there is a large accuracy loss.
>
> **A2**: Thanks for your consideration. Our proposed technique can be applied to both fine-tuning and pre-training, and we run experiments on both fine-tuning and pre-training tasks. Even though there is still an accuracy loss on pre-training tasks, our result for pre-training is on par with previous 4-bit training methods for dedicated hardware [6, 7], whose titles are also training rather than fine-tuning.
>
> In general, 4-bit training for transformers is still an extremely challenging task to study. We believe that our work makes a reasonable step towards both 4-bit fine-tuning and pre-training, despite there is still room to improve.
>
>
> **Q3**: The speed-up of up to 35.1\% is misleading. It is better to use the average speed-up in the abstract.
>
> **A3**: Thanks for the constructive comment. We apologize for any confusion this may have caused.
> We will use the average speed-up in the final version, as suggested by the reviewer.
>
> We are grateful for the time and effort you have invested in evaluating our submission. We are more than willing to address any further questions or concerns you may have.
>
> The references are listed in the global rebuttal section.

---

> > ### Comment · Reviewer_jhrg · 2023-08-17
> > **Reply**
> >
> > Most of my concerns are addressed. Thanks!

---

### Author Rebuttal · Authors · 2023-08-10

Due to the length limit, we have included the references we utilized in the general rebuttal section.

[1] Zhu, Feng, et al. "Towards unified int8 training for convolutional neural network." Proceedings of the IEEE/CVF Conference on Computer Vision and Pattern Recognition. 2020.

[2] Zhao, Kang, et al. "Distribution adaptive int8 quantization for training cnns." Proceedings of the AAAI Conference on Artificial Intelligence. Vol. 35. No. 4. 2021.

[3] Zhang, Xishan, et al. "Fixed-point back-propagation training." Proceedings of the IEEE/CVF conference on computer vision and pattern recognition. 2020.

[4] Wang, Shuai, and Yi Kang. "Gradient distribution-aware INT8 training for neural networks." Neurocomputing 541 (2023): 126269.

[5] Zhou, Qihua, et al. "Octo:{INT8} training with loss-aware compensation and backward quantization for tiny on-device learning." 2021 USENIX Annual Technical Conference (USENIX ATC 21). 2021.

[6] Sun, Xiao, et al. "Ultra-low precision 4-bit training of deep neural networks." Advances in Neural Information Processing Systems 33 (2020): 1796-1807.

[7] Chmiel, Brian, et al. "Accurate Neural Training with 4-bit Matrix Multiplications at Standard Formats." The Eleventh International Conference on Learning Representations. 2022.

[8] Dettmers, Tim, et al. "Qlora: Efficient finetuning of quantized llms." arXiv preprint arXiv:2305.14314 (2023).

[9] Wortsman, Mitchell, et al. "Stable and low-precision training for large-scale vision-language models." arXiv preprint arXiv:2304.13013 (2023).

[10] Banner, Ron, et al. "Scalable methods for 8-bit training of neural networks." Advances in neural information processing systems 31 (2018).

[11] Wang, Naigang, et al. "Training deep neural networks with 8-bit floating point numbers." Advances in neural information processing systems 31 (2018).

---

### Decision · Program_Chairs · 2023-09-21

**Decision:**

Accept (poster)

**Comment:**

This paper proposes methods to accelerate the training of transformer models by utilizing 4-bit integers and arithmetic. The proposed methods include a Hadamard quantizer for forward propagation to suppress the outliers. In the backward propagation, the paper employs a bit splitting scheme to exploit the sparsity of gradients. The proposed methods are evaluated on transformer models such as BERT and ViT, demonstrating competitive accuracy and speedup on GPUs.

Overall, the paper is well-written. The techniques introduced are innovative, substantiated by a theoretical analysis. The evaluation is reasonably thorough. The proposed algorithm does not necessitate specialized hardware, enabling its implementation on standard GPUs.

The main concern is on the evaluation of hardware performance. Given the current INT4 algorithms introduce noticeable overhead in order to recover the accuracy loss, it aligns with the reviewers’ perspective that a comparison with simple INT8 training is necessary to understand the advantages of the algorithms. It is understandable that a comprehensive demonstration of the speedup is complex and challenging; however, a high-level complexity analysis and the inference speed with I-BERT, as presented in the rebuttal, should be incorporated in the final version with detailed explanation.

In summary, this paper is technically solid with certain reservations. The authors have, to some extent, addressed the concerns in the rebuttal phase. Consequently, it is recommended that the paper be considered for borderline acceptance.